# Simultaneous functional MRI of two awake marmosets

Kyle M. Gilbert [1,2✉], Justine C. Cléry [1], Joseph S. Gati[1,2], Yuki Hori[1], Kevin D. Johnston [3,4], Alexander Mashkovtsev[1], Janahan Selvanayagam [1,5], Peter Zeman[1], Ravi S. Menon [1,2], David J. Schaeffer[6] & Stefan Everling[1,3]

Social cognition is a dynamic process that requires the perception and integration of a complex set of idiosyncratic features between interacting conspecifics. Here we present a method for simultaneously measuring the whole-brain activation of two socially interacting marmoset monkeys using functional magnetic resonance imaging. MRI hardware (a radio-frequency coil and peripheral devices) and image-processing pipelines were developed to assess brain responses to socialization, both on an intra-brain and inter-brain level. Notably, the brain activation of a marmoset when viewing a second marmoset in-person versus when viewing a pre-recorded video of the same marmoset—i.e., when either capable or incapable of socially interacting with a visible conspecific—demonstrates increased activation in the face-patch network. This method enables a wide range of possibilities for potentially studying social function and dysfunction in a non-human primate model.

[1] Centre for Functional and Metabolic Mapping, The University of Western Ontario, London, ON, Canada. [2] Department of Medical Biophysics, The University of Western Ontario, London, ON, Canada. [3] Department of Physiology and Pharmacology, The University of Western Ontario, London, ON, Canada. [4] Brain and Mind Institute, The University of Western Ontario, London, ON, Canada. [5] Graduate Program in Neuroscience, The University of Western Ontario, London, ON, Canada. [6] Department of Neurobiology, University of Pittsburgh, Pittsburgh, PA, USA. ✉email: kgilbert@robarts.ca

Social cognition is a dynamic process that requires the perception and integration of a complex set of idiosyncratic features between interacting conspecifics. Although investigations pairing unimodal stimuli with functional imaging have yielded major insights into the neural correlates of social interaction, the idiosyncrasies embedded in real social interactions—such as those communicated by reactive facial expression—are lost in these highly controlled paradigms[1]. Clever implementations of functional magnetic resonance imaging (fMRI) involving subjects interacting over a network, called hyperscanning[2–4] has been employed where two people are simultaneously scanned in disparate scanners that are connected through an audio–video link. Hyperscanning is particularly useful for studying the unpredictability of social interactions, whereby participants' behaviours are impacted by each other[5–7]. It has been noted, however, that brain activation is increased when subjects have a truly live interconnection versus watching a recorded interaction[8]. To remove the confounds of studying virtual interactions, radio-frequency (RF) coils—the hardware components responsible for receiving the MRI signal from the brain—have been developed that allow for the simultaneous imaging of two people within the same MRI scanner[9–11]. Although an elegant solution, these studies are inhibited by the limited space within the bore, which requires subjects to be in close physical contact and therefore create an unnatural social dynamic, particularly between two unrelated subjects.

Social interaction has likewise been studied in preclinical animal models, which enables the use of multi-modal and electrophysiological measurements to assess brain activation. Correlation and synchrony of neural activity during social interaction has been demonstrated in mice, bats, and non-human primates: calcium imaging of socially interacting mice has demonstrated synchrony of their neural activity predictive of social behaviour[12]; wireless electrophysiology used to record local field potentials of socially interacting bats has demonstrated correlation of neural activity over a range of timescales[13]; while neuronal ensemble recordings of non-human primates have shown inter-brain cortical synchronization during social interaction[14]. In fact, the mere presence of another monkey during the completion of a task has been demonstrated to increase brain activity in the attention frontoparietal network using FDG-PET[15].

Extending these animal models to represent neuropsychiatric disorders, such as schizophrenia, depression, and bipolar disorder, remains a challenging field of study[16]; however, the common marmoset monkey (*Callithrix jacchus*) is emerging as a popular animal model due to its close homology with humans in comparison to rodents[17–20] and due to its granular dorsolateral prefrontal cortex[21], a region of the brain that has been linked to a variety of neuropsychiatric disorders and social cognition[22–24]. Marmosets also exhibit prosocial and cooperative behaviour, akin to humans, thus promoting their use as a model of social behaviour[25]. This small, New-World primate, reaches sexual maturity quickly and has a high birth rate, making it an ideal candidate for transgenic studies[26]. Marmosets can be trained to perform complex behavioural tasks while head-fixed[27], allowing them to be used to study brain function while awake[28–32]. Recently, MRI of marmosets in the fully awake state has been performed to eliminate the confounds of anaesthesia on functional activation. This technique requires specialized RF coils, such as conformal designs that clamp an individual marmoset's head[33–35], restraint devices with built-in RF coils[36], and RF coils with integrated clamps to fixate an implanted chamber[37].

Although simultaneous anatomical MRI studies have been conducted of animals (in particular, mice) for genetic studies[38,39], the study of socially interacting animals has yet to be investigated with fMRI—a technique which would allow a whole-brain assessment of activation (in contrast to higher spatial-resolution electrophysiological recordings and calcium imaging) in multiple animals simultaneously.

In this work, we describe a method (referred herein as the "social-coil method") wherein hardware (including an RF coil and positioning platform) and image-processing pipelines are developed to enable simultaneous fMRI of two socially interacting marmosets on a clinical MRI scanner. The salient metrics of the coil topology are evaluated, in the context of the unique technical challenges attributed to a dual-marmoset design, to address the primary question: can the requisite image quality be achieved (in terms of signal-to-noise ratio (SNR) and limited image distortion) to map intra-brain neuronal activity and inter-brain activity of socially interacting marmosets? In the first experiment, the method's efficacy is demonstrated by measuring the intra- and inter-brain activation of two marmosets who are continually within each other's visual field. In the second experiment, social interaction is demonstrated by comparing the brain activation of a marmoset when either viewing a second marmoset in-person or when viewing a pre-recorded video of the same marmoset—i.e., when either capable or incapable of socially interacting with a visible conspecific. The method described in this manuscript allows for the marmoset to be adopted as an animal model to investigate whole-brain activation during social interaction in primates, enabling the study of the neural basis of social deficits in neuropsychiatric disorders using transgenic marmosets.

## Results

**Design of the radiofrequency-coil system.** The radiofrequency-coil system was designed to achieve three primary goals: (1) to mitigate animal motion during functional scanning; (2) to allow marmosets to have reproducible and variant orientations within the scanner; and, most importantly, (3) to produce the requisite sensitivity for mapping brain activation on a 3 T MRI scanner.

The RF coil was comprised of two disparate receive coils: each coil consisting of a marmoset restraint system with an integrated radiofrequency array. This topology was adapted from our previously published design for imaging awake-behaving marmosets on a 9.4 T small-animal scanner[37]. As the limited homogeneous region of the 9.4 T scanner precluded studying two marmoset brains, even in close proximity, modifications were made to allow use of our previous design on a 3 T whole-body scanner and permit flexibility in the mechanical setup. The restraint system consisted of an acrylic tube equipped with neck and tail plates to constrain body motion (Fig. 1). The RF coil was affixed to the inner surface of an integrated head-fixation system, wherein the act of closing the two halves of the hinged RF coil would clamp an implanted head chamber[37,40] while simultaneously electrically completing the coil element circumscribing the chamber. The close-fitting nature of the receive array increases sensitivity and therefore image quality; four-point fixation of the chamber minimized translational and rotational motion to less than 140 μm and 0.6° over a 5-min functional run (Supplementary Fig. 1). Motion was considerably smaller than the 1-mm voxel size implemented in this study, thereby resulting in a negligible effect on data quality.

Each receiver coil was comprised of 5 elements tuned to the Larmor frequency of protons within the 3 T scanner: 123.2 MHz. Preamplifiers used in this study had the ubiquitous $B_0$ orientation-dependence caused by the Hall effect[41]. To prevent deleterious changes to the preamplifier noise figure, and hence image SNR, long coaxial cables were used to attach preamplifiers

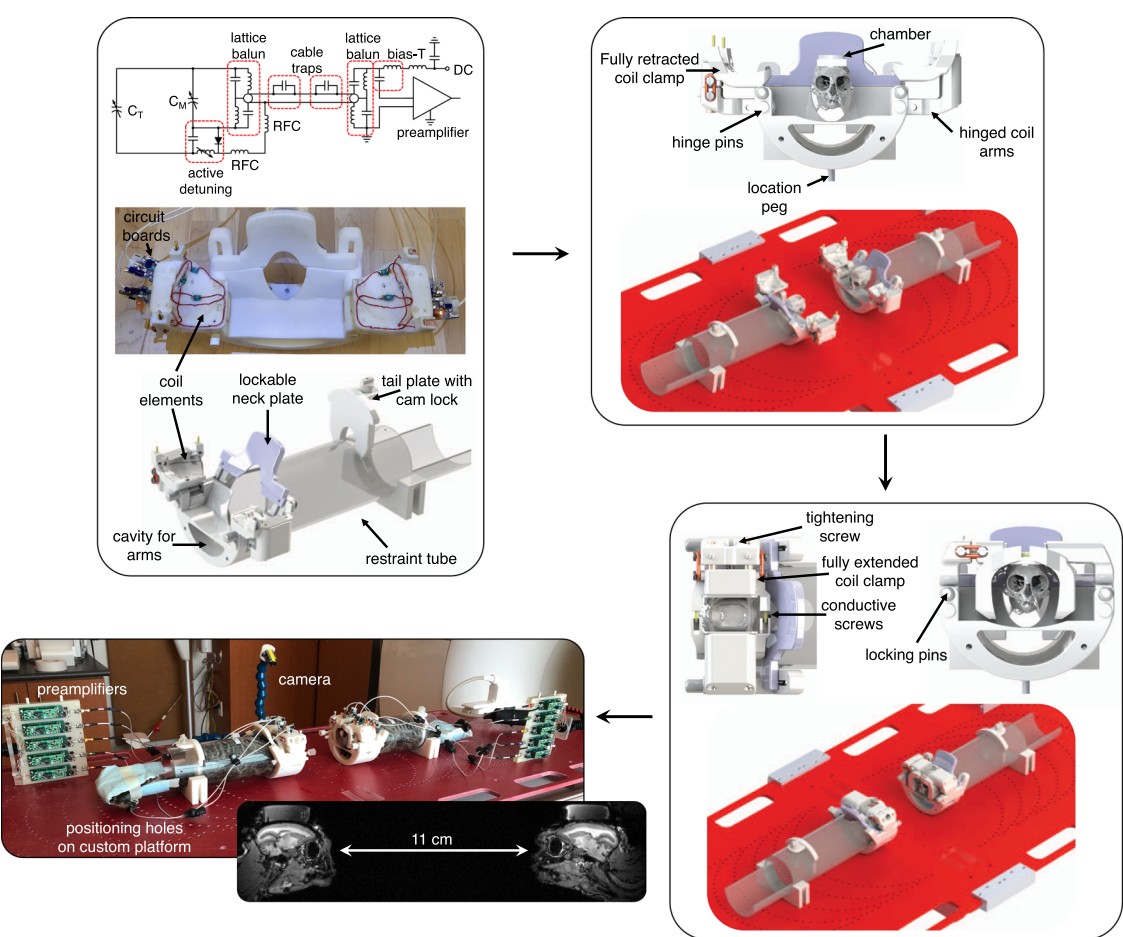

**Fig. 1 Mechanical setup of the social coil.** Each marmoset is placed in a restraint device with an integrated radiofrequency coil. Initially, the arms of the restraint device are fully opened to allow the marmoset to enter a tube and be restrained by lockable neck and tail plates. Coil elements are located on the inner surface of two hinged arms; circuit boards interfacing to the coil elements are adhered to the top of the hinged arms. Once the marmosets' bodies are restrained, they are placed on a custom platform that allows reproducible and variant positioning within the scanner. Gel is subsequently placed on the marmosets' heads to reduce susceptibility artifacts and geometric distortion. Marmosets are then head-fixed by closing the hinged coil arms, inserting the locking pins, and fully extending the coil clamp with a tightening screw. This creates a four-point fixation of the chamber and electrically completes the coil element circumscribing the chamber by pressing conductive screws into opposing conductive pads. Preamplifier modules are repositioned independent of the coil to ensure low-noise amplifiers always maintain their optimal orientation with respect to the main magnetic field regardless of coil rotation. In this study, marmosets were placed 11-cm apart and facing each other: a distance chosen to allow natural social interaction based on the visual acuity of the marmoset. A camera was employed to monitor marmosets during scanning. $C_T$: tuning capacitor; $C_M$: matching capacitor; RFC: radiofrequency choke; DC: direct current.

to the coil: these enabled preamplifiers to be mounted to modules that were independent of the coil housing, allowing them to maintain the correct orientation with respect to $\mathbf{B_0}$ regardless of coil position. The electrical schematic of a single receive element is provided in Fig. 1.

A dedicated platform was constructed to allow reproducible and variant positioning of the two marmoset coils (Fig. 1). Two pegs underneath the coil housing could be inserted into an array of holes in the platform allowing the coil to be rotated about the $y$-axis of the scanner and translated along the $z$-axis. The confluence of the allowable translation and rotation allowed the marmosets to be set up anywhere from facing each other (allowing direct eye contact) to entirely parallel to each other (allowing both marmosets to view a common mirror or projector screen, or potentially additional animals).

**Evaluating noise characteristics of the social coil.** Similarity between the performance metrics of the disparate receive coils is imperative to facilitate unbiased comparisons of brain activation between marmosets (i.e., to ensure measured differences in brain

connectivity are physiological and not caused by differing coil performance). To this end, geometric decoupling, preamplifier decoupling, and active detuning were measured on the bench when coils were loaded with tissue-mimicking phantoms.

The mean and maximum coupling between receive elements was −16 dB and −12 dB, respectively (coil 1) and −22 dB and −12 dB, respectively (coil 2). The low-input-impedance preamplifiers achieved a mean decoupling per coil of −24 dB (worst-case: −20 dB), resulting in a mean and maximum in vivo noise correlation (Fig. 2a) of 12% and 28%, respectively (coil 1) and 13% and 29%, respectively (coil 2). The electrical and physical separation between the two coils ensures they are intrinsically well decoupled: the maximum inter-coil coupling was 2.3%. The low correlation coefficient between coils ensures that any observed synchronous brain activity between marmosets is not an artifact due to inter-coil coupling. The difference in mean noise level between coils was 9.7%, indicating similar noise characteristics were achieved through construction. Active detuning provided a minimum isolation of −29 dB between the transmit and receive coils during transmission.

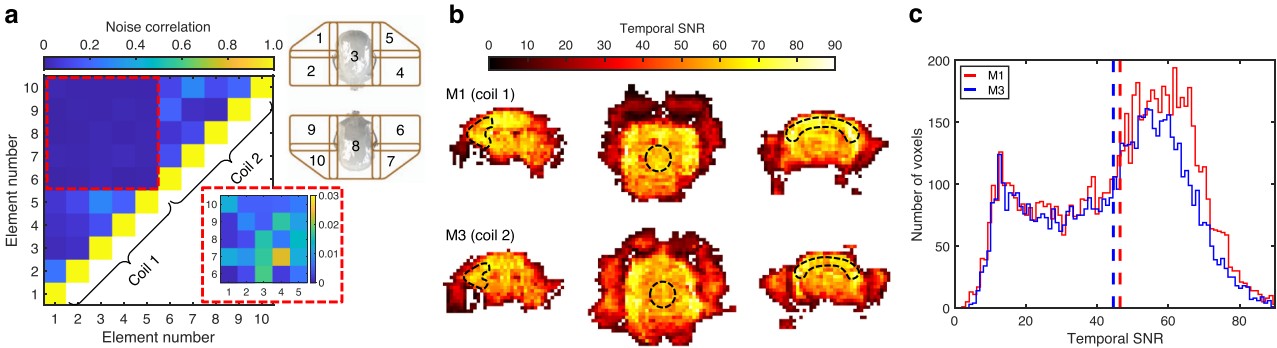

**Fig. 2 Signal and noise characteristics of the social coil. a** Intra- and inter-coil noise correlation of the two receive arrays corresponding to the coil-element layout and numbering depicted in the upper inset (represented in a planar view: coil 1: elements 1–5; coil 2: elements 6–10). The lower inset refers to inter-coil noise correlation, which has a maximum of 2.3%. **b** Temporal SNR of marmoset M1 (coil 1) and marmoset M3 (coil 2) when face-to-face along the z-axis of the scanner. ROIs (dashed regions) in the sagittal, axial, and coronal planes show inter-brain regional differences of less than 10%, which is less than intra-brain differences (which are approximately 2 to 3-fold). Maps have been reoriented into radiological convention to facilitate comparison. **c** Histograms of the temporal SNR of each voxel within the brain of each marmoset show similar distributions. Dashed lines represent the mean temporal SNR for each marmoset/coil combination.

**Evaluating temporal signal-to-noise ratio for functional imaging**. Dedicated ultra-high-field small-animal scanners can provide high $B_0$ fields that allow for increased SNR and image resolution[42–44]. The drawback of such systems is the typical reduction in bore size and therefore a limitation on imaging volume. Clinical scanners[45], in contrast, can accommodate larger RF coils with more peripheral equipment and have larger imaging volumes: this can be exploited to scan multiple marmosets simultaneously, albeit at the expense of SNR (due to the lower field strength).

The temporal SNR must be high enough to accommodate a sufficient resolution to discriminate the spatial origins of the BOLD fMRI signal: this is challenging on a clinical scanner owing to the small subject size, yet large field-of-view required to accommodate two marmosets. Secondly, the two RF coils must produce consonant temporal SNR profiles to mitigate the confound of spatially varying sensitivity when quantifying synchronous brain activation.

Temporal SNR maps, derived from a 1-mm-isotropic spatial resolution echo-planar-imaging (EPI) time course, were acquired of two marmosets facing each other (Fig. 2b, c). The mean temporal SNR over the brain of each marmoset was 46.4 and 44.5, respectively—values sufficient for network mapping, as demonstrated in proceeding sections. The difference in temporal SNR between marmosets when averaged over the whole brain was 4%, whereas regional differences amounted to 9% in the frontal lobe, 7% in the centre of the brain, and 6% in the peripheral motor cortex. These small discrepancies in temporal SNR between marmosets can be attributed to both minor differences in the two RF coils as well as anatomical differences between the two marmosets.

Intra-brain heterogeneity of the temporal SNR profiles is intrinsic to all surface-coil arrays and is approximately 2 to 3-fold for the social coil. Heterogeneity in sensitivity profiles, both intra-brain and inter-brain, produce commensurate heterogeneity in the spatial sensitivity to neural correlation. To improve the accuracy of inter-brain connectivity analyses, the power of functional connectivity maps should be greater than the variability in inter-brain heterogeneity in temporal SNR, allowing for direct comparison of brain activation between interacting marmosets.

**Correcting geometric distortions on a clinical scanner**. Imaging two disparate subjects within a single imaging volume reduces the efficacy of $B_0$ shimming, with the result being a potential increase in geometric image distortion for susceptibility weighted protocols with long echo trains, such as is utilized with gradient-echo EPI. This problem is exacerbated when trying to $B_0$ shim over a small volume (i.e., the marmoset brain) with a clinical human whole-body gradient/shim coil; however, the lower field strength in relation to ultra-high-field small-animal scanners produces a commensurate reduction in local field inhomogeneities caused by differences in magnetic susceptibility, such as near the sinuses, the base of the skull, and regions surrounding the chamber.

No significant difference in image distortion was discerned when $B_0$ shimming over both marmoset brains simultaneously versus over one marmoset brain at a time. Image distortion was predominantly localized to the temporal poles and the region where the chamber was affixed to the skull. This susceptibility boundary caused local off-resonance fields of up to 225 Hz, as determined with successive runs with opposing phase-encode direction[46]. This off-resonance field map was subsequently applied to correct for geometric distortion. Residual distortion was sufficiently minimal to allow accurate registration of functional images to an anatomical template[47] (Fig. 3). The local field fidelity therefore met the threshold for reducing image distortion to a manageable level, despite the challenges of $B_0$ shimming over two disparate and small regions-of-interest on a clinical 3 T scanner.

**Intra-brain network mapping with a visible conspecific**. Functional runs with a continually visible conspecific were acquired to determine whether the temporal SNR and image resolution (1-mm isotropic) would be sufficient to map intra-brain connectivity. Four runs, each 10 min long, were acquired with two marmosets placed face-to-face and 11-cm apart—this allowed for marmosets to be within each other's visual field for the entire run. Independent component analysis was applied to each brain to determine intra-brain correlations during constant social interaction. Intra-brain connectivity maps (z-score maps) of three representative networks are presented in Fig. 4 (see Supplementary Fig. 2 for all derived network maps). Seven statistically significant networks were found: three somatosensory networks (SMNs), a default mode network (DMN), a primary visual network (VISp), a high-order visual network (VISh), and a salience network (SAN)—networks previously confirmed to be present in the resting marmoset[28,37,48].

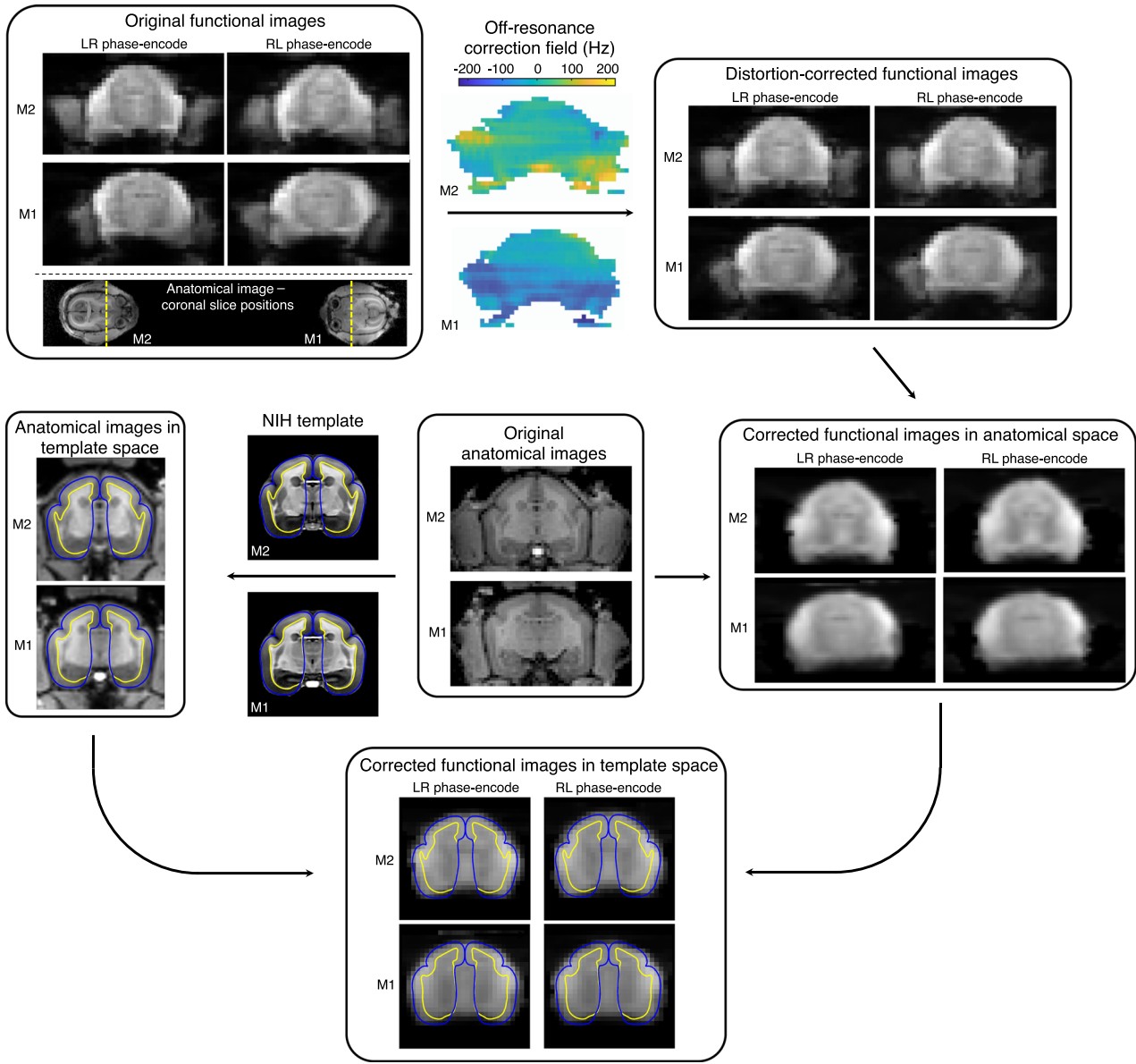

**Fig. 3 Image distortion due to local magnetic field inhomogeneities.** The unique dual-marmoset setup has implications on image distortion during fMRI acquisitions: face-to-face marmosets experience opposing phase-encode directions, which creates different geometric distortion. To compensate, echo-planar images are acquired with alternating phase-encode directions. In this topology, marmoset M1, with left-right (LR) phase encode, will have similar distortion to marmoset M2 with right-left (RL) phase-encode. The most notable image distortion is found in the temporal poles and at the boundary of the chamber. Successive functional runs are then used to estimate the off-resonance correction field required for correcting image distortion. After functional images are distortion-correction and brain-extracted, they are registered to an anatomical image (i.e., anatomical space). Anatomical images, having been registered to the NIH marmoset brain atlas[67], are used to register the functional images (in anatomical space) onto the brain atlas (template space). Registration of functional images to the marmoset brain atlas facilitates the assignment of localized brain activation to known networks. Grey- and white-matter boundaries are shown as blue and yellow lines, respectively.

**Inter-brain synchronous connectivity with a visible conspecific.** One of the unique advantages to scanning multiple awake marmosets simultaneously is the ability to assess synchronous (or conversely time-lagged and/or asynchronous) whole-brain connectivity between animals. To demonstrate the efficacy of the social-coil method, functional time-courses acquired with a continually visible conspecific were assessed for inter-brain synchrony. The z-score, derived from the correlation coefficient between time courses of spatially analogous voxels, showed correlated (synchronous) activity in regions A13M, A23, A24, AuCL, and V1 (Fig. 5). Of note, A24 is a region of the brain known to be involved in social-interaction processing in primates[49]. Variations

in task design, marmoset pairings, and behaviour would likely evoke different regions of synchronous activity and could therefore be tailored to the neuroscientific question of interest with respect to social function and dysfunction.

**Assessing socialization with in-person versus pre-recorded conspecific.** Brain activation maps of a marmoset were acquired when either viewing a second marmoset in-person (paradigm 1) or when viewing a pre-recorded video of the second marmoset (paradigm 2)—i.e., when either capable or incapable of socially interacting with a visible conspecific. In each of the two experiments, two marmosets (the second being in-person or a video)

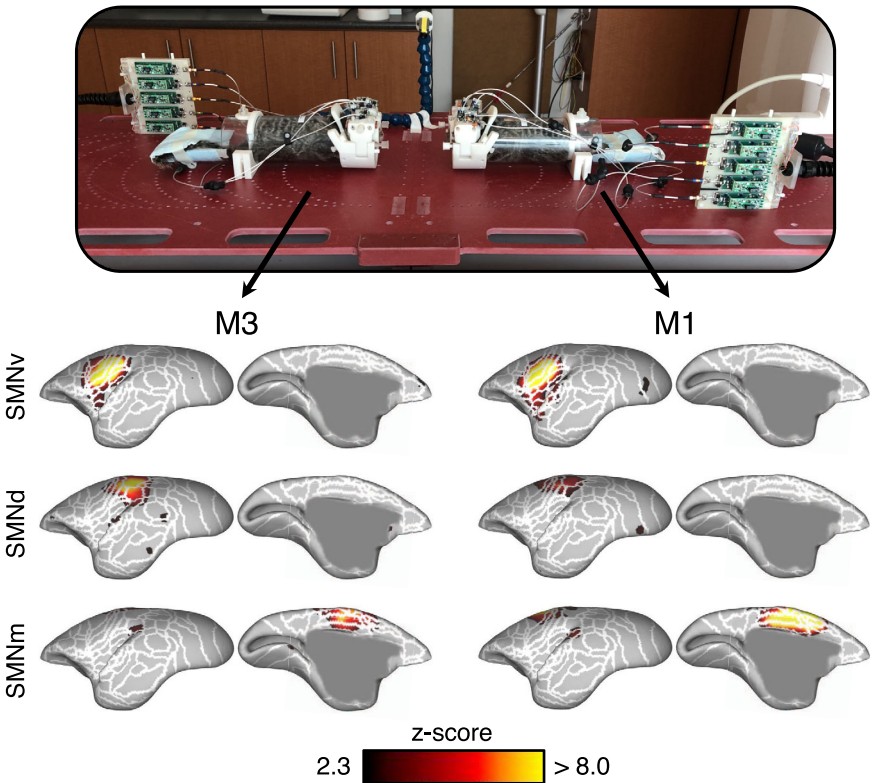

**Fig. 4 Intra-brain activation maps derived from the simultaneous scanning of two marmosets within each other's visual field.** The social coil provided sufficient temporal SNR and resolution to discern seven functional networks in marmosets M1 and M3. Three representative functional networks are displayed: the ventral somatomotor network (SMNv), dorsal SMN (SMNd), and medial SMN (SMNm). Networks are displayed as z-score maps on the cortical surface, with only the left hemisphere visible. White lines represent cytoarchitectonic borders. Supplementary Fig. 1 shows all remaining networks in both hemispheres and at the volume level.

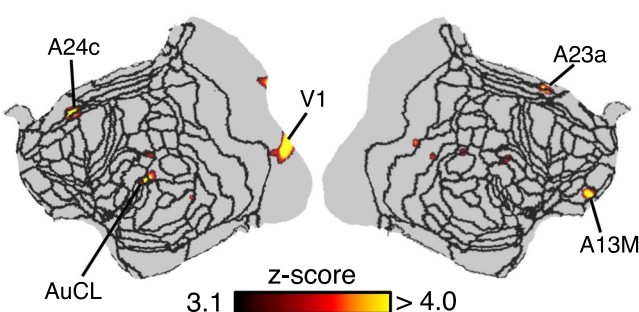

**Fig. 5 Synchronous brain activation between two marmosets within each other's visual field.** Two marmosets, M1 and M3, were placed facing each other while acquiring four functional time courses. The correlation coefficient between the time courses of spatially analogous voxels in marmosets M1 and M3 was calculated and transformed to a z-score map—z-scores are presented on a flattened map of the left and right hemispheres, with black lines indicating cytoarchitectonic borders. Synchronous activity was found in the anterior cingulate cortex, ventromedial prefrontal cortex, and temporal cortex—regions thought to play an important role in social interaction.

were placed face-to-face and separated by smart films, the opacity of which was alternated in a block design (Fig. 6a) to permit or deny visual contact; regions of increased brain activation were derived in each experiment by comparing the two epochs of the block design. For each paradigm, the stimulus condition (transparent smart films) was compared to the baseline condition (opaque smart films) using a two-sided paired t-test (Fig. 6b, c);

an unpaired t-test between the two paradigms' stimulus conditions discerned regions of the brain preferentially activated when the marmoset was viewing another marmoset (face-to-face versus face-to-video; Fig. 6d).

Activation patterns derived from paradigms 1 and 2 had similar spatial distributions, with significant activation occurring for both paradigms in multiple cortical areas (Fig. 6b): visual (V1, V2, V3, V4, V4T, V6A, V6, 19 M), temporal (IPa, TE3, temporo-parieto-occipital association, fundus of the superior temporal, middle superior temporal, middle temporal), parietal (PFG, PG, PGM, occipito-parietal transitional area of cortex, anterior intraparietal, lateral intraparietal, medial intraparietal, ventral intraparietal), frontal (dorsorostral and dorsocaudal parts of area 6, dorsal and ventral parts of area 8, caudal part of area 8, part a and b of the ventral area 6, part c of the primary motor area 4), and cingulate (23a, 24a, 24b, 24c, 24d, 25) cortex. At the subcortical level, significant activation was found across both paradigms in the putamen, caudate, amygdala, thalamus, superior colliculus, and cerebellum (Fig. 6c).

Viewing a second marmoset in-person (paradigm 1) compared to a video (paradigm 2) resulted in significantly higher activations in regions corresponding to previously described temporal (AD, MD, PV, and PD) and frontal (area 8b/24) face patches[22,50]. In addition, we observed higher activations in the right motor and premotor areas (area 4, area 6DC, are 6DR, 8 C) (Fig. 6d).

## Discussion

Investigating the brain's reaction to socialization, using fMRI, relies on the amalgamation of dedicated hardware with image-processing pipelines. This manuscript describes a set of such tools that facilitate the simultaneous fMRI of two marmosets.

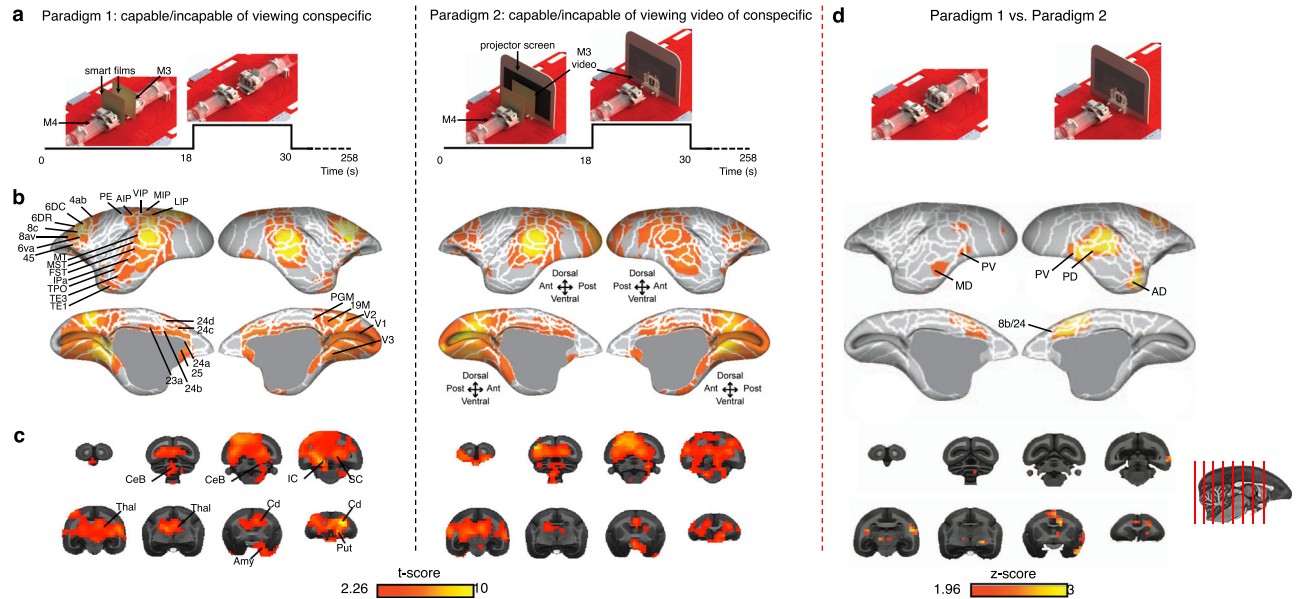

**Fig. 6 Preferentially activated brain regions in two marmosets within each other's visual field. a** Two task-based fMRI paradigms were acquired. In paradigm 1, a marmoset (M4) could view a second marmoset (M3) in-person—i.e., it was capable of socially interacting with a visible conspecific. In paradigm 2, M4 viewed a pre-recorded video of M3—i.e., it was incapable of socially interacting with a visible conspecific. In each paradigm, the two marmosets (M3 being in-person or a video) were placed face-to-face, 11-cm apart, and separated by smart films. The opacity of the smart films was alternated in a block design to permit or deny visual contact. **b** Regions of increased brain activation were derived for each paradigm by comparing the two epochs of the block design in **a** using a two-sided $t$-test. Activation maps are represented on the left and right fiducial surface of the M4 marmoset brain ($t$-scores >2.26, $p < 0.05$, uncorrected for multiple comparisons), with white lines representing cytoarchitectonic borders. **c** The equivalent activation maps as represented on coronal slices to highlight subcortical activations. **d** A voxel-wise, two-sided $t$-test between the stimulus conditions of paradigms 1 and 2 indicates regions of increased activation due to socialization ($z$-scores > 1.96, $p < 0.05$, uncorrected for multiple comparisons). The displayed sagittal slice indicates the position of coronal slices.

The radiofrequency coil was designed to minimize animal motion during fully awake imaging sessions, produce high sensitivity for connectivity mapping, and have the flexibility to support varying physical setups within the scanner. The confluence of these design characteristics permits the simultaneous measurement of whole-brain activation in two marmosets, enabling a variety of interactions to be investigated, from direct eye contact to the viewing of a common screen. The open-chamber design extends the flexibility of the method by allowing the integration of complimentary modalities, such as electrophysiology, to augment fMRI studies.

Operating on a 3 T clinical scanner allowed for a large enough imaging region to simultaneously scan two marmosets. The caveat to using a clinical gradient coil optimized for human use is the commensurate reduction in gradient performance (i.e., rise time and maximum gradient strength). The consequent increase in echo spacing leads to greater image distortion in sub-optimally $B_0$-shimmed ROIs; however, local susceptibility gradients in the magnetic field (and therefore geometric image distortion) were sufficiently mitigated to allow accurate registration of functional images to an anatomical template with delineation between grey and white brain matter—a requisite for discriminating the spatial origins of the BOLD signal and for group analysis of multiple marmosets.

The social coil hardware and optimized imaging protocol provides sufficient temporal SNR to produce maps of intra-brain activation and inter-brain synchronicity at a 1-mm-isotropic resolution. The seven intra-brain functional networks obtained while continually in the presence of a visible conspecific were in agreement with resting-state networks previously observed using ultra-high-field MRI[37]. Despite the lower spatial resolution compared to typical ultra-high-field MRI studies (around 0.5-mm isotropic), the well-described small activation in the dorsolateral

frontal area in DMN[51,52] was detected. Nonetheless, increasing the number of coil elements would facilitate higher acceleration rates, resulting in reduced geometric image distortions and permitting higher resolution functional data—an achievable goal due to the high temporal SNR.

Notably, the social-coil method was capable of revealing synchronous neuronal activity between two marmosets. Synchronicity was found in regions thought to play an important role in social interaction (the anterior cingulate cortex (ACC), ventromedial prefrontal cortex (mPFC), and temporal regions). For example, these brain regions in macaque monkeys have shown increased activity (using BOLD-based functional MRI) when they watch movies showing monkeys interacting versus acting independently[49]. In humans, mutual interaction during eye contact has been shown to be mediated by the limbic mirror system, including the ACC and anterior insular cortex (AIC)[53]. Furthermore, inter-brain synchrony has been found in regions including the ACC and right AIC when playing a modified interactive ultimatum game[54]. Taken together, correlated activity in area 24 (a part of the anterior and middle cingulate cortex) and the temporal region infers a synchronization of regional brain activity due to the presence of a visual conspecific. The ability for the social-coil method to detect synchronous activity allows for the study of increasingly sophisticated animal models and social behaviour.

Brain-activation maps generated of a marmoset when viewing a second marmoset in-person or when viewing a pre-recorded video of the second marmoset showed activation networks previously identified in marmosets performing tasks[30,32,55]—most notably, this activation was found in regions associated with visuomotor tasks. An in-person conspecific resulted in significantly higher activations in the anterior cingulate (area 8B, area 24) and temporal regions. These areas correspond to previously identified

frontal[22] and temporal face patches in marmosets[22,50], supporting a role of the face patch network in social cognition[56]. Overall, differences between the brain-activation maps of the two paradigms are consistent with behavioural evidence of mutual gaze (see Supplementary Methods and Discussion). In humans, it has recently been suggested that eye contact during conversation indicates shared attention[57].

The method described in this study enables the measurement and assessment of synchronous neuronal activation, across the whole brain, between marmosets. In-person interaction between marmosets is shown to produce localized increases in brain activity compared to the presence of a visible, but non-interacting, marmoset—demonstrating the efficacy of the method in evaluating social interaction. This allows the study of social function and dysfunction in a non-human primate model, including the use of transgenic models of neuropsychiatric disorders. The demonstration of simultaneous functional imaging of two marmosets within the same scanner removed the confounds of remote hyperscanning. The modular coil design can now be expanded to allow for the simultaneous scanning of larger cohorts, enabling a broad range of social groups and interactions to be investigated.

## Methods

**Social-coil hardware topology**. The social coil was comprised of a marmoset restraint system with an integrated radiofrequency coil (Fig. 1). This system was adapted from our previously published design for imaging awake-behaving marmosets on a 9.4 T small-animal scanner[37].

The body-restraint consisted of a 7-cm-diameter acrylic tube (product number: 8532K23; McMaster-Carr, Aurora, Ohio, USA) with neck and tail plates. Marmosets would enter the tube, be restrained by an adjustable neck plate with thumb screws and a tail plate with cam lock: the neck plate and receive coil were angled to accommodate the marmoset's body. The marmoset would then lie in the sphinx position with the coil arms fully opened. Once acclimatized to being in the body-restraint, marmosets would be head-fixed by closing the coil arms and locking them in place by inserting hinge pins. The retractable clamp is then tightened, by turning a screw, to secure the fixation pins into the chamber. The head-fixation assembly was 3D-printed using stereolithography with a photopolymer resin (White Resin V4; Form 2, Formlabs, Somerville, Massachusetts, USA).

**Design and fabrication of the integrated receive array**. The receive array included five loops integrated into the interior of the coil former to allow close proximity to the brain (and therefore higher SNR). One element circumscribed the head chamber, while two elements were arranged along the anterior–posterior direction on either side of the head: this allowed for a two-fold acceleration rate in the anterior–posterior or left–right directions during parallel imaging. Coil elements were geometrically overlapped to reduce inter-element coil coupling[58]. Each element was constructed from 22-AWG insulated copper wire and included three or four distributed capacitors to reduce conservative electric fields: one variable capacitor for matching (Sprague-Goodman Electronics, SGC3 series), one fixed capacitor for the active-detuning circuit, and one variable capacitor for tuning (an additional surface-mount capacitor was incorporated into the large element circumscribing the chamber). Each element was tuned to 123.2 MHz and matched to $200 + j50\ \Omega$ (i.e., the optimal noise impedance of the preamplifier) when loaded with a 2.5-cm-diameter spherical phantom filled with 50-mM sodium chloride. Polyurethane foam (product number: 86375K162; McMaster-Carr, Aurora, Ohio, USA), 1.6-mm-thick, was adhered to the inner surface of the coil as a spacer to prevent conservative electric fields from coupling to the marmoset (which would result in a shift in the resonant frequency and a reduction in SNR).

The element circumscribing the chamber was split into two halves. When tightening the retractable clamp to secure the chamber, two conducting posts (6–32 brass screws) on one arm pushed into two conductive pads on the other arm to create electrical continuity. The conductive pads were created by wrapping copper braid in front of ethylene-vinyl-acetate (EVA) foam and soldering the braid to one half of the coil element. The flexibility of the foam created solid connections between the conducting posts and associated pads to decrease coil noise and prevent spiking artifacts.

Circuit boards were mounted on the outside of the former and included a matching capacitor, an active-detuning circuit, and a lattice balun. The low-input-impedance preamplifiers (Stark Contrast, Erlangen, Germany) had a $B_0$ orientation-dependence due to the Hall effect. To prevent a reduction in preamplifier noise figure, 61-cm-long RG178 coaxial cables (i.e., a half-wavelength electrical length) attached the coil to the preamplifiers through non-magnetic MCX connectors. Preamplifiers were then mounted to modules that were independent of

the coil housing, yet movable themselves, and consistently orientated with the low-noise amplifier parallel to $B_0$. The longer cables required additional cable traps to suppress common-mode currents: in addition to the lattice baluns placed at the input of the coil, a lattice balun was located at the input of the preamplifier (to ensure a symmetric input to the preamplifier) and two choke baluns were placed along the coaxial cable. The second lattice balun had the opposite orientation to the lattice balun at the coil input to counteract any impedance transformation. The baluns were sufficient at preventing changes in the tune and match of coil elements with varying cable position. The half-lambda cables transformed the low input impedance of the preamplifier to a parallel-resonant inductance across the matching circuit, creating a high-impedance circuit to reduce inter-element coupling (i.e., preamplifier decoupling[58]). A multi-conductor cable connected preamplifiers to the system socket; this cable had a split sleeve balun to reduce common-mode coupling to the transmit coil.

**Design and fabrication of a custom positioning platform**. A dedicated platform was constructed to allow reproducible positioning of the two marmoset coils (Fig. 1). The platform was comprised of 0.95-cm-thick garolite with legs that slotted onto the scanner bed. Slots were machined on the side of the platform to allow for ease of handling.

Two pegs underneath the coil housing could be inserted into an array of holes in the platform: each coil could be rotated about the $y$-axis of the scanner by up to 180°, in 5° increments; the $z$-position of each coil could be varied in 5-cm increments to allow approximately 1–91 cm of space between marmosets. Marmosets could be oriented to allow face-to-face interaction with direct eye contact to being entirely parallel to each other to view a common mirror or projector screen. Since the sensitivity to transverse magnetization decreases with increased angle to $B_0$, coils should be placed at conjugate angles with respect to $B_0$ to avoid introducing an SNR disparity between the two coils. A quantitative measurement and explanation of this effect is provided in the Supplementary Methods and Discussion.

**Bench evaluation of the social coil**. Standard single- and double-probe techniques[59] were employed to measure geometric decoupling, preamplifier decoupling, and active detuning on the bench using a network analyzer. Geometric decoupling was measured as the transmission coefficient between preamplifier ports. Preamplifier decoupling and active detuning were measured as the difference between the tuned state (i.e., without a preamplifier present or detuning bias) and when the coil had a preamplifier present and was tuned (preamplifier decoupling) or detuned (active detuning). Coils were loaded with 2.5-cm-diameter spherical phantoms filled with a 50-mM sodium-chloride solution to approximate physiological conductivity.

**The marmoset animal model**. Experimental procedures were in accordance with the Canadian Council of Animal Care policy and a protocol (#2017-114) approved by the Animal Care Committee of the University of Western Ontario Council on Animal Care. Imaging was performed on four common marmosets (named M1, M2, M3, and M4): 3-year-old males weighing 310 g (M1), 400 g (M2), and 340 g (M3), and a 2.5-year-old female weighing 365 g (M4). Marmosets M1 and M3 (scanned for network mapping with a visible conspecific) were housed together and are twin brothers. Marmosets M3 and M4 (scanned for assessing socialization with in-person versus a pre-recorded visible conspecific) were housed separately and not familiar with each other. Benchtop eye-tracking experiments were performed between marmoset M4 and M5 (a 4-year-old female weighing 490 g). Marmosets M4 and M5 were housed separately and not familiar with each other.

Marmosets underwent an aseptic surgical procedure[40] to implant a head chamber while the animal was placed in a stereotactic frame (Narishige, Model SR-6C-HT). Adhesive resin (All-bond Universal, Bisco, Schaumburg, Illinois, USA) was applied using a microbrush, air dried, and cured with an ultraviolet dental curing light (King Dental), after which a two-component dental cement (C & B Cement, Bisco, Schaumburg, Illinois, USA) was applied to the skull and the bottom of the chamber. The chamber was then lowered onto the skull with a stereotactic manipulator to ensure accurate placement. A 3D printed cap was attached to the chamber with set screws; this cap was removed before entering the magnet room.

To minimize stress and anxiety during scanning, prior to the first imaging session marmosets were trained for three weeks following an established acclimatization procedure[37,60]. In week 1, marmosets were constrained in the restraint tube (Fig. 1)—without head fixation—for durations increasing up to a maximum of 30 min. In week 2, the restraint tube was inserted into a mock MRI tube and gradient-coil sounds were played at increasing volume (up to ~80 dB) and for durations increasing up to a maximum of 60 min. In week 3, marmosets were head-fixed with the fixation pins (Fig. 1), inserted into the mock MRI tube, and exposed to MRI sounds.

During each training session, animals were rewarded with pudding or marshmallow fluff for remaining calm, facing forward, and having minimal limb movement. Marmosets' tolerance to training was assessed using a behavioural rating scale[60]: each marmoset was required to score a 1 or 2 on the assessment scale prior to scanning.

Directly prior to scanning, marmosets were placed in the restraint system, but not head fixed, in a preparation room adjacent to the magnet room. Marmosets were head-fixed once on the scanner bed to minimize risk during transfer. During imaging, an MRI-compatible camera was used by a veterinary technician to monitor marmosets for stress and to check as to whether the animals were awake.

Marmosets did not vocalize with each other[61,62] while head-fixed in the social coil; although head-fixation inhibits vocal communication, it is nonetheless a requisite to successfully perform MRI. Marmosets did, however, exhibit behaviour indicative of social interaction—i.e., mutual eye gaze. This was validated through simultaneous eye tracking of two head-fixed marmosets while on the bench (see Supplementary Methods and Discussion).

**MRI scanner hardware**. All imaging was performed at the Centre for Functional and Metabolic Mapping at The University of Western Ontario. MRI data collection was performed on a human, whole-body Siemens Prisma Fit scanner (Siemens Healthineers AG, Erlangen, Germany) operating with a 3-T main magnetic field. The system is equipped with 64 receiver channels, of which 10 were utilized: two plugs (one per coil) were interfaced to Tim coil interface adaptors to allow operation on the Siemens Prisma hardware platform. The XR80/200 gradient coil allowed for a maximum gradient strength of 80 mT/m and a maximum slew rate: 200 T/m/s.

Whether an MRI scanner is capable of supporting this method is dependent on the size of the gradient coil: so as long as both marmosets can be placed within the linear region of the gradient coil, and sufficiently distant for their visual acuity, this technique can be translated to different field strengths and MRI scanners.

**Measuring the temporal signal-to-noise ratio**. A gradient-recalled-echo, noise-only acquisition (i.e., without RF transmission) was acquired to calculate the noise level and noise correlation matrix of and between the two disparate coils (matrix size: $384 \times 156$, FOV: $256 \times 104$ mm, TE/TR: 4.6/10 ms, BW: 260 Hz/pixel).

Temporal SNR maps were calculated from a single-shot, EPI time series with two marmosets (M1 and M3) facing each other: FOV: $220 \times 78$ mm, acquisition matrix: $220 \times 78$, number of slices: 25, slice thickness: 1 mm, TE/TR: 30/1,500 ms, BW: 1,265 Hz/pixel, flip angle: 70°, volumes: 400, acceleration rate: 2 (right-left), reference lines: 24, partial Fourier encoding: 6/8. Temporal SNR maps were calculated by measuring the ratio of the mean signal of each pixel through the detrended time course to the standard deviation of that pixel through the time course. Temporal SNR calculations were performed in Matlab (The MathWorks, Natick, MA).

ImageJ was used to calculate the mean temporal SNR in regions of interest located in the frontal lobe, centre of the brain, and peripheral motor cortex. The mean temporal SNR was also calculated over the entire brain region (i.e., excluding the temporal muscles, eyes, etc.).

Spatial variations in SNR due to flip-angle inhomogeneity were found to be minimal, as described in the Supplementary Methods and Discussion.

**Supplementary coil performance metrics**. Image SNR, receive sensitivity, and geometry factor were measured as supplementary metrics to assess coil performance. A full description of this analysis is provided in the Supplementary Methods and Discussion.

**Assessing geometric distortion**. Head chambers often produce local image-intensity dropouts due to differences in the magnetic susceptibility between the chamber, air, and tissue. Susceptibility-induced distortion attributed to the chamber was partially abated by applying a water-based gel (MUKO SM321N; Canadian Custom Packaging Company, Toronto, Canada) to the brow ridge and inside the chamber, thus reducing susceptibility differences close to the brain.

Marmosets M1 and M2 were placed facing each other 11-cm apart—a distance chosen based on the visual acuity of marmosets[63]. $B_0$ shimming was performed over a volume large enough to encompass both marmoset brains. Two single-shot, multiband[64] EPI functional runs were acquired of both marmosets, simultaneously during a single session (orientation: axial; FOV: $220 \times 78$ mm, acquisition matrix: $220 \times 78$, number of slices: 25, slice thickness: 1 mm, TE/TR: 30/1,500 ms, BW: 1265 Hz/pixel, flip angle: 70°, acceleration rate: 2, reference lines: 24, partial Fourier encoding: 6/8). Accelerated images were reconstructed with generalized autocalibrating partially parallel acquisition (GRAPPA)[65].

Since marmosets have opposite orientations within the scanner, their respective image distortions, caused by local field inhomogeneities, have opposing anatomical directions. To correct for this difference, successive functional datasets were acquired with opposite phase-encode direction (left–right versus right–left).

From these successive runs, the susceptibility-induced off-resonance field was estimated[46] and used to correct for distortion (FSL[66]; topup). A magnetization-prepared, rapid gradient echo (MPRAGE) image was acquired as an anatomical reference (FOV: $220 \times 68$; matrix: $352 \times 110$, number of slices: 64, resolution: 0.63-mm isotropic, TE/TI/TR: 4.2/900/2300 ms, BW: 200 Hz/pixel, flip angle: 9°, number of averages: 2). Functional images were registered to the NIH marmoset brain atlas[67] as described in the proceeding section.

**Intra-brain network mapping with a visible conspecific**. Functional time courses (echo-planar images) were acquired of two marmosets while they were continually within each other's visual field. Four runs (during a single session), with 400 volumes each, were acquired with marmoset M1 and M3 facing each other and placed 11-cm apart. Pulse-sequence parameters were identical to those described in the previous section, with alternating phase-encode direction along the left–right axis (i.e., left-to-right versus right-to-left).

Prior to functional runs, $B_0$ shimming was performed over the imaging volume encompassing both marmosets. An anatomical reference scan (the MPRAGE sequence) was acquired of the multiple marmosets for image registration (pulse-sequence parameters were identical to those described in the previous section). Functional data was pre-processed and analyzed using an adaptation of our previously published method[37], as described below.

Anatomical images were split into separate datasets for each marmoset. These datasets were then reoriented to radiological convention to compensate for the different orientation of each marmoset within the scanner. The process of removing the skull from the anatomical image was conducted in three stages: (1) the olfactory bulb was manually removed, as it was not included in the template image; (2) the delineation of the brain–skull boundary was approximated (FSL; BET; radius: 25–30 mm; fractional intensity threshold: 0.3); and (3) the $T_1$-weighted brain template[67] was linearly and nonlinearly registered to the skull-stripped anatomical image (FSL; FLIRT and FNIRT) to refine the brain-skull boundary, create a mask, and create an atlas-to-anatomical transformation matrix.

Functional images were split to have one distinct dataset per marmoset. Data from each marmoset was preprocessed separately using FSL[66]. As with anatomical images, functional images were reoriented to the standard radiological convention. Functional images were then corrected for motion (FSL; FLIRT) and geometric distortion (FSL; topup) and manually brain extracted (FSL; fslview). An average functional image was calculated for each run and an anatomical-to-functional transformation matrix was calculated (FSL; FLIRT and FNIRT).

Functional images (still in native space) were finally normalized to the template using the inverse of the transformation matrices (functional-to-anatomical and anatomical-to-atlas)—this facilitated the assignment of brain activation to known brain regions. This was followed by spatial smoothing (1.5-mm full-width at half-maximum Gaussian kernel) and temporal filtering (0.01 to 0.1 Hz) (FSL; fslmaths).

Principal component analysis (PCA) was applied to remove the unstructured noise from each animal's time course (FSL; MELODIC), followed by independent component analysis (ICA) with 100 dimensions. The resultant components were classified as signal or noise based on the criteria as shown in previous reports[68]. Noise components were regressed from each fMRI time course (FSL; fsl_regfilt). Group ICA, with 20 dimensions, was subsequently performed on each marmoset's data to detect the neural components.

**Inter-brain synchronous connectivity with a visible conspecific**. Inter-brain synchronous connectivity was deduced by calculating the voxel-wise correlation coefficient between the simultaneously acquired functional time series of the two marmosets that were continually within each other's visual field. After time-course data was preprocessed, as described in the previous section, voxel intensities were subsequently normalized to have a mean of zero and a standard deviation of 1. (This latter step mitigated bias due to inter-run differences in the mean and standard deviation of voxel intensities.) Time courses of the four independent runs were temporally concatenated on a voxel-wise basis. The correlation coefficient was then calculated between spatially analogous voxels in the disparate marmoset brains. Correlation-coefficient maps were Fisher-Z transformed and masked to a threshold of 3.1. Calculations were performed in Matlab (The MathWorks, Natick, MA).

**Assessing socialization with in-person versus pre-recorded visible conspecific**. A marmoset's brain activation was measured when either viewing another marmoset in-person (paradigm 1) or when viewing a pre-recorded video of the same marmoset (paradigm 2).

In paradigm 1, two marmosets (M3 and M4) were placed face-to-face, at a distance of 11 cm. Smart films (ASIN: B077P4QJT1; HOHOFILM) were secured approximately 2 cm in front of each marmoset. In a block paradigm, the opacity of the smart films was alternated by the application of a voltage: each run was comprised of 17 alternating blocks—18 s in the opaque condition (i.e., no visible conspecific) followed by 12 s in the transparent condition (i.e., capable of interacting with a visible conspecific). (Smart films were located close to the marmosets' faces to prevent them from focusing on potential reflections off of the screen.) The voltage was controlled by a Raspberry-Pi (Raspberry-Pi 3, Model B), through a Python script, which was synced to the trigger output of the scanner that was sent at the beginning of each volume acquisition.

In paradigm 2, M3 was replaced by a pre-recorded video of itself projected onto a screen located 11 cm in front of M4. The video consisted of a recording of M3 while head-fixed in the receive coil and on the positioning platform. The brightness of the video display was adjusted to best replicate the light conditions of the face-to-face paradigm. The opacity of the smart films was once again alternated to switch between the two conditions: no visible conspecific versus a visible conspecific without the ability to socially interact.

Functional (EPI) and anatomical (MPRAGE) data were acquired with the same acquisition parameters as described for the experiment with a continuous visible conspecific, albeit with 172 volumes per run. Each experiment consisted of 2 sessions, with 4 or 6 runs per session: the 10 total runs consisted of 5 runs with left–right acceleration and 5 runs with right–left acceleration.

Task-based data was pre-processed using primarily the same pipeline as described in the previous section, with small differences pertaining to the processing of functional images. After image reorientation, motion correction, and distortion correction, functional images (in native space) were despiked (AFNI; 3dDespike) and volume registered to the middle volume of each time series (AFNI; 3dvolreg). Images were smoothed by a 2.0-mm full-width at half-maximum Gaussian kernel to reduce noise (AFNI; 3dmerge).

The two paradigms—face-to-face and face-to-video—were analyzed independently. For each paradigm, task timing was convolved with the hemodynamic response function (AFNI; BLOCK convolution), and for each run a regressor was generated for each condition (i.e., transparent smart films (stimuli) versus opaque smart films (baseline)) to be used in a regression analysis (AFNI; 3dDeconvolve). Both conditions were entered into the same model, along with fifth-order-polynomial detrending regressors, bandpass regressors, and the motion parameters derived from the volume registration. The resultant regression-coefficient maps were then registered to the NIH template space[67] using the transformation matrices described in the previous section.

A direct comparison between the two paradigms was conducted at the individual level by performing an unpaired, voxel-wise t-test between the two paradigms' stimulus conditions (AFNI; 3dttest ++)—i.e., face-to-face with transparent smart films versus face-to-video with transparent smart films. Resultant z-values were displayed on fiducial maps using Connectome Workbench[69] in conjunction with the NIH marmoset brain template[67].

To ensure the switching of the smart film did not alter the received signal, scans were acquired without marmosets (i.e., noise-only images) using the identical paradigm—no difference in noise level was observed between states of the smart film, nor were noise spikes observed during the switching of the smart film.

**Assessing motion during functional runs.** Head motion was estimated for marmosets M1 and M3 during the four, 10-min functional runs acquired for assessing intra-brain network mapping with a continual visible conspecific. Each functional volume was registered to the middle volume of its respective time course (FSL; mcflirt_acc). Time-varying motion parameters—translation (left–right, inferior–superior, posterior–anterior) and rotation (pitch, roll, yaw)—were then extracted for each run.

**Functional data acquisitions.** EPI datasets acquired in this study employed imaging volumes large enough to encompass both marmosets, which requires the acquisition of non-encoded space and therefore increased noise. This, however, is a limitation imposed by the pulse sequence and is a tractable problem. Multi-animal functional MRI would lend itself well to simultaneous multi-slice imaging techniques[70]. With the modification to allow for two stacks of slices, a slice acceleration factor of two-fold would allow for simultaneous acquisition of a single slice in each marmoset acquired from disparate, decoupled RF coils. Consequently, the distance between marmosets could be increased without a proportional increase in imaging volume, and commensurately the SNR should improve.

**Reporting summary.** Further information on research design is available in the Nature Research Reporting Summary linked to this article.

## Data availability
The raw and preprocessed functional and anatomical data generated in this study have been deposited in a public OSF repository[71] (https://doi.org/10.17605/OSF.IO/EJGF8). Raw data has been provided for Figs. 2–6 and Supplementary Figs. 1, 2, 5, and 6.

## Code availability
Computer-aided design files, image-processing pipelines, a Python V3.4.2 acquisition script, and a custom Matlab R2018b analysis tool have been provided in a public OSF repository[71] (https://doi.org/10.17605/OSF.IO/EJGF8).

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

## Acknowledgements

The authors would like to thank Cheryl Vander Tuin and Hannah Pettypiece for animal preparation and care and Scott Charlton for scanning assistance. This work was supported by the Canada Foundation for Innovation (to RSM); Canada First Research Excellence Fund to BrainsCAN; Brain Canada Platform Support Grant (to RSM); Natural Sciences and Engineering Research Council of Canada Discovery Grant (to RSM); and Canadian Institutes of Health Research FRN 148365 (to SE) and FRN 353372 (to RSM); BrainsCAN Postdoctoral Fellowship and Canadian Institutes of Health Research Post-doctoral Fellowship FRN 176571 (to JCC).

## Author contributions

K.M.G., D.J.S., R.S.M., and S.E. conceived the social-coil method. K.M.G. and D.J.S. designed the restraint system. K.M.G. and A.M. fabricated the radiofrequency coil and acquired bench measurements. P.Z. designed and fabricated the custom platform. K.M.G., J.C.C., Y.H., J.S.G, and S.E. designed and conducted experiments. K.M.G. analyzed coil-performance data. J.C.C. and Y.H. analyzed functional data. K.D.J., J.S., and S.E. performed and analyzed eye tracking data. K.M.G., J.C.C., Y.H., and D.J.S. wrote the manuscript. All authors reviewed the manuscript.

## Competing interests

The authors declare no competing interests.
