## [Peer Review File · Nature Communications]

Simultaneous functional MRI of two awake marmosetsREVIEWER COMMENTS

Reviewer #1 (Remarks to the Author):

This is a fantastic study done by Gilbert and colleagues. Their work is opening a new avenue of research for social neurosciences.

The authors claimed that their new social coil could allow for marmosets to have reproducible and variant orientations within the scanner. How does the flexibility in coil configuration / different orientations of the social coil impact on signal quality?

Compared to previous studies, the DMN recorded with this new coil is more restricted (Liu et al. 2019 Nat Comm; Hori et al. 2020 Cereb Cortex). For instance, no activity in the medial prefrontal cortex is observed with the social coil recordings compared to recordings collected with a more standard setup. The lateral frontal activity is also lacking in one animal and reduced compared to the one reported by Hori and colleagues. Is this a potential limitation of the social coil or the fact that a 3T MRI scanner was employed instead of a 7T or 9.1T machine?

Fig6: the authors might want to refer to area 13m as ventromedial prefrontal cortex instead of medial prefrontal cortex

Reviewer #2 (Remarks to the Author):

The article describes a method for simultaneous fMRI scanning of two marmosets using a regular clinical 3T MR scanner. It can unveil the changes produced by an experimental condition within the brain of each animal as well as their correlation and synchrony -or lack thereof - across the two subjects.

This represents a groundbreaking innovation in the neuroimaging literature. The common marmoset (*Callithrix jacchus*), a small primate, is currently emerging as a powerful model for the future of neuroscience. One of the reasons is its quick reproduction and the successful generation of transgenic marmosets as mentioned by the authors. Regarding social cognition, it has been argued in addition that the marmoset shares with humans prosocial and cooperative behaviors and could be, in this respect, a better animal model of human social behavior than the more despotic rhesus macaque, the historical primate model in neuroscience (Miller et al. 2016. Marmosets: A Neuroscientific Model of Human Social Behavior. *Neuron*, 90(2), 219–233). Marmosets can be relatively easily bred and 3T clinical scanners are relatively common worldwide, so the new approach described in the paper is highly valuable to the neuroscience community and has the potential to be extensively disseminated across the world, and to be especially helpful in the social cognition domain. I am not an fMRI expert but as far as I can judge, the methods and analyses are sound, and clearly reported so that others can apply the method on their own.

Minor concerns.

Animal welfare is an important societal issue today. A brief description of how stress and anxiety were prevented during restraint training would probably be welcome.

Four marmosets were included in the study. But it is not clear which pairs were scanned for which analyses and for how many runs and sessions. e.g. Page 25: Marmosets M1 and M2 were placed facing each other; Figure 6: Two marmosets, M1 and M3, were placed facing each other; Figure 5 and Page 28: Two marmosets (M2 and M4) were placed face-to-face.

Familiarity modulates social influences. Were the marmosets familiar with each other? Housed together?

The authors may want to add some results about estimated motion (mean, or derivative motion regressors) so that the reader can get an idea of the direction and amplitude of the movements

allowed by the restraint system.

Major concern

I do not think that the behavior measured qualifies as a "social interaction", let alone a "constant social interaction". The bore was alternatively blacked-out and illuminated during scanning. So brain activation was measured simultaneously in two animals facing either a visible or invisible conspecific. There was an online video control of whether or not the marmoset was awake, but no offline quantification of the time actually spent looking at the conspecific. Social interaction is the reciprocal influence subjects exert on each other. The two marmosets in the present study exert no influence on each other, and may or may not look at each other. So the affirmation repeated throughout the manuscript that the data provides a brain network mapping of constant social interaction should probably be tempered accordingly. What is detected is more modestly the brain network associated with a visible conspecific.

Martine Meunier

Reviewer #3 (Remarks to the Author):

This study describes a non-invasive fMRI method for studying brain activity in two marmosets simultaneously during 'social interaction' in a human 3T scanner. There have been previous studies of social interaction using simultaneous scanning (hyperscanning) of 2 or more people in single (e.g. dyads: Miyata et al Neuroimage Mar 2021, triads: Xie et al 2020 PNAS 117:23066) and multiple scanners (e.g. Read Montague). Multiple mice have been scanned in single MRI scanners, but not in the context of social interaction. To my knowledge, there are no publications of direct social interactions between animals in single scanners. This study scans two marmosets simultaneously as they view each other face to face. Marmosets live in families, exhibit characteristic social behaviors, are known especially for their auditory processing (Xiaoqing Wang), and social calls (Asif Ghazanfar). They have also become a subject of neuroscience studies, transgenic manipulation, and models of diseases such as autism (e.g. Mimura 2019 Neuroimage 195:243). Marmosets have thus become a popular nonhuman primate model of human behavior and disease. The combination of a strong nonhuman primate model and hyperscanning is thus a quite novel approach. The authors have developed a cutting edge technical advance in overcoming the many challenges of field inhomogeneity, RF coil design, and characterizing and decoupling noise in the two coils. This engineering is impressive and sets the stage for a new paradigm of studies in primate models of social interaction. While the authors are to be commended for such an advance, with respect to social interactions, it is not yet clear that this setup can be used for studying social interactions in a way that is more 'social' than viewing another marmoset on a screen. In my opinion, this is a great proof-of-principle that this approach can be achieved technically and I recognize what a tour-de-force effort this is. However, as presented, it may be more suitable for a journal targeted to new technologies or methods.

Major comments

1. To make this a more scientific question-based study, it would be helpful to offer some hypothesis or some predictions about which circuits social interaction would be activated by in-person interaction vs more passive monitor-based presentations (something quite relevant in this COVID world!). The data could then be probed to address such hypotheses.
2. Social Interaction face-to-face vs viewing on monitor: Perhaps the most pertinent question is whether and to what extent the marmosets were socially interacting. Comparison of brain activity during viewing of another marmoset face on a monitor would be a reasonable comparison. Some controls would be helpful. These could take the form of devices to track the animal's physiological and behavioral state, neurophysiological signals that correlate with attention (e.g. pupil size or other signatures), monitoring of the animals face and ears, tracking hormonal levels.
3. Social interaction vs vision: The primary control here is turning out the lights, during which it is assumed that the marmosets do not 'see' each other, thereby reducing the amount of social interaction. This comparison could also be interpreted as simply presence vs absence of visual inputs. Therefore, visual stimulus activations should also be presented to demonstrate that the differences observed are beyond simple vision.
4. Social interaction vs gender: Another useful comparison might be comparison of same sex and opposite sex brain activations. One would predict different activations based on gender.

5. Marmoset behavior is strongly dominated by auditory inputs and their interpretation. Does the loud scanner noise lead to impairing of social interaction or to stress response?
6. Social activity is typically accompanied by strong amygdala activation, as has been demonstrated in humans and macaque monkeys. Was this observed in the marmoset scans?
7. References are not complete.

Minor comments

1. Could this be done in a 9.4T? To what extent could this be conducted with commercial coils? Could you provide spatial SNR, g-factor, and the B1 maps/efficiency of two marmoset brains that are 11cm away from each other.
2. To highlight the advantages of this approach, some comparison of this study with those of e.g. multichannel electrophysiological recordings and of miniature scope imaging would be useful. What are the similarities and differences in results and in approach?

Anna W. Roe, Zhejiang University

Response to Reviews

We would like to thank the reviewers and editor for taking the time to provide thoughtful feedback on our manuscript. We have made amendments to the manuscript to address these concerns and are excited about the improvements. Most substantially, we have replaced our second experiment with one that compares the brain activation of a marmoset when either viewing a second marmoset in-person or when viewing a pre-recorded video of the same marmoset—i.e., when either capable or incapable of socially interacting with a visible conspecific. When viewing a second marmoset in-person, activation was increased in regions associated with social cognition, thereby demonstrating that the marmosets are indeed interacting.

We have also included additional data for coil sensitivity as a function of angle, transmit-field uniformity, image SNR, receive sensitivity, geometry factor, and motion estimates.

A Supplementary Information file has been included with additional Supplementary Methods, Discussion, and Figures. Changes to this Supplementary document have not been annotated in order to conform to the formatting requirements of the journal.

Additional minor modifications have been highlighted within the manuscript. The editorial policy checklist, reporting summary, and software policy have been updated. Data and code have been updated and moved to a public OSF repository ([DOI: 10.17605/OSF.IO/EJGF8](https://doi.org/10.17605/OSF.IO/EJGF8)).

Reviewer #1

This is a fantastic study done by Gilbert and colleagues. Their work is opening a new avenue of research for social neurosciences.

Thank you for taking the time to review our manuscript and for your enthusiasm for our work. We have made amendments to our paper in response to each of your comments, as detailed below.

R1.1. The authors claimed that their new social coil could allow for marmosets to have reproducible and variant orientations within the scanner. How does the flexibility in coil configuration / different orientations of the social coil impact on signal quality?

The angle of the receive elements with respect to B_0 will affect sensitivity, since the B_1^- in the transverse plane decreases with an increased angle to B_0 ; therefore, we recommend having both coils at the same angle with respect to B_0 . We mentioned this briefly in the Methods section of the original submission, but we have now augmented this with an experiment. We rotated the coil through a range of angles, θ , with respect to the z-axis of the scanner and measured the covariance-weighted, root-sum-of-squares SNR of a spherical phantom. B_1^- (and SNR) followed the expected cosine dependency with angle to B_0 . From Gauss's Law, the divergence of a magnetic field is zero (i.e., $\nabla \cdot \mathbf{B}_1 = 0$); therefore, even when the normal vector of the coil element is parallel to B_0 , it will still have a transverse component (i.e., a non-zero sensitivity). However, as θ increases, the four lateral elements of the coil produce less transverse B_1^- and the combined sensitivity profile of

the coil becomes dominated by the coil element at the superior aspect of the head, which is always orthogonal to B_0 . The greatest decrease in image SNR, 18%, occurred at an angle of approximately 75°. The minimum does not occur at 90° because the normal vectors of the side elements are not at right angles to the longitudinal axis of the coil.

This new data has been added to the Supplementary Methods and Discussion. A Supplementary Figure has been added with a plot of the mean image SNR as a function of θ , and with the corresponding central transverse slice of corresponding image SNR maps.

Supplementary Methods:

“**Assessing coil sensitivity dependence on angular position.** The social coil is amenable to variable positioning within the scanner, including altering the coil’s angle with B_0 . Since the coil’s sensitivity to transverse magnetization is related to the transverse component of the receive field, B_1^- , the angle of the coil with respect to B_0 will alter the SNR profile. To quantify this dependency, a coil was loaded with a 3.8-cm-diameter spherical phantom, filled with 50-mM sodium chloride, and rotated through a range of angles with respect to B_0 (0° to 90° in 5° increments). At each angle, a 3D gradient-recalled echo was acquired with and without RF transmission (matrix size: 224 × 92 × 64, FOV: 179 × 73 × 51.2 mm, TE/TR: 4.6/10 ms, flip angle: 20°, BW: 220 Hz/pixel, number of averages: 2). The covariance-weighted, root-sum-of-squares SNR was calculated^{1,2} at each angle using the complex data from individual elements.”

Supplementary Discussion:

“Image SNR followed the expected cosine dependency with angle to B_0 (Supplementary Fig. 3). As the angle increases, the four lateral elements produce less transverse B_1^- and the coil element at the superior aspect of the head (which remains orthogonal to B_0) begins to dominate the combined sensitivity profile. The largest decrease in image SNR, 18%, occurs at approximately 75°: the minimum does not occur at 90° because the normal vectors of the lateral elements are not orthogonal to the longitudinal axis of the coil. It is therefore recommended to position the two coils at conjugate angles to minimize their SNR disparity.”

“Supplementary Fig. 3 | Coil sensitivity dependence on angular position. The social coil can be positioned at varying angle to the longitudinal axis of the scanner to allow different physical arrangements for social interaction. Coil sensitivity to transverse magnetization (and therefore image SNR) is dependent on the angle of each element with B_0 . Mean image SNR, as a function of the coil positioning angle, is presented with the corresponding central transverse slice of the image SNR map. Lateral elements produce diminishing transverse B_1 with increased angle; however, from Gauss’s Law, the divergence of a magnetic field is zero (i.e., $\nabla \cdot \mathbf{B}_1 = 0$); therefore, even when the normal vector of a coil element is parallel to B_0 , it will still have a transverse component (i.e., a non-zero sensitivity).”

R1.2. Compared to previous studies, the DMN recorded with this new coil is more restricted (Liu et al. 2019 Nat Comm; Hori et al. 2020 Cereb Cortex). For instance, no activity in the medial prefrontal cortex is observed with the social coil recordings compared to recordings collected with a more standard setup. The lateral frontal activity is also lacking in one animal and reduced compared to the one reported by Hori and colleagues. Is this a potential limitation of the social coil or the fact that a 3T MRI scanner was employed instead of a 7T or 9.1T machine?

This is an issue of field strength and the power of the maps, rather than a limitation of the social coil. The temporal SNR is reduced at 3T compared to 7T or 9.4T, so more runs would be required to attain a clearer DMN. In Hori et al., 4 monkeys were utilized, with 6 runs, each 15-min long,

per monkey. In this study, 4 monkeys were utilized, with 4 runs, each 5-min long, per monkey. A statement has been added to the caption of Supplementary Fig. 2 to clarify this point.

“Improvements in z-score and network mapping can be achieved by averaging additional functional runs.”

R1.3. Fig6: the authors might want to refer to area 13m as ventromedial prefrontal cortex instead of medial prefrontal cortex.

We have now changed the description to “ventromedial prefrontal cortex” in the figure and in the main body of the manuscript.

Reviewer #2

☒ **The article describes a method for simultaneous fMRI scanning of two marmosets using a regular clinical 3T MR scanner. It can unveil the changes produced by an experimental condition within the brain of each animal as well as their correlation and synchrony -or lack thereof – across the two subjects.**

This represents a groundbreaking innovation in the neuroimaging literature. The common marmoset (*Callithrix jacchus*), a small primate, is currently emerging as a powerful model for the future of neuroscience. One of the reason is its quick reproduction and the successful generation of transgenic marmosets as mentioned by the authors. Regarding social cognition, it has been argued in addition that the marmoset shares with humans prosocial and cooperative behaviors and could be, in this respect, a better animal model of human social behavior than the more despotic rhesus macaque, the historical primate model in neuroscience (Miller et al. 2016. *Marmosets: A Neuroscientific Model of Human Social Behavior*. *Neuron*, 90(2), 219–233). Marmosets can be relatively easily bred and 3T clinical scanners are relatively common worldwide, so the new approach described in the paper is highly valuable to the neuroscience community and has the potential to be extensively disseminated across the world, and to be especially helpful in the social cognition domain. I am not an fMRI expert but as far as I can judge, the methods and analyses are sound, and clearly reported so that others can apply the method on their own.

Dr. Meunier,

Thank you very much for taking the time to provide helpful feedback on our manuscript and for your enthusiasm for our study. In light of your major concern as to whether the experiments illustrate social interaction versus the visualization of a conspecific, we have made two major changes to the manuscript. In the first experiment (“Constant” social interaction), we have now tempered the language to align with your comment. We have now replaced the entire second experiment (Intermittent social interaction) with an experiment that we believe demonstrates that the animals are indeed interacting: this experiment compares the brain activation of a marmoset when either viewing a second marmoset in-person or when viewing a pre-recorded video of the same marmoset—i.e., when either capable or incapable of socially interacting with a visible conspecific. We have also added a citation to Miller et al. in regard to social behaviour, as you describe. Detailed responses to individual comments, including a description of the new experiment, are provided below.

Minor concerns.

R2.1. Animal welfare is an important societal issue today. A brief description of how stress and anxiety were prevented during restraint training would probably be welcome.

We have now expanded our description of the restraint-training procedure in the Methods section, which includes an assessment and reduction of animal stress.

“To minimize stress and anxiety during scanning, prior to the first imaging session marmosets were trained for three weeks following an established acclimatization procedure^{3,4}. In week 1, marmosets were constrained in the restraint tube (Figure 1)—without head fixation—for durations increasing up to a maximum of 30 min. In week 2, the restraint tube was inserted into a mock MRI tube and gradient-coil sounds were played at increasing volume (up to ~80 dB) and for durations increasing up to a maximum of 60 min. In week 3, marmosets were head-fixed with the fixation pins (Figure 1), inserted into the mock MRI tube, and exposed to MRI sounds. In the current setup, auditory interaction between marmosets^{5,6} is inhibited by the loud noises of the MRI scanner; although this may prove to be a tractable problem through engineering, it presently limits this form of social interaction.

During each training session, animals were rewarded with pudding or marshmallow fluff for remaining calm, facing forward, and having minimal limb movement. Marmosets’ tolerance to training was assessed using a behavioral rating scale³: each marmoset was required to score a 1 or 2 on the assessment scale prior to scanning.”

R2.2. Four marmosets were included in the study. But it is not clear which pairs were scanned for which analyses and for how many runs and sessions. e.g. Page 25: Marmosets M1 and M2 were placed facing each other; Figure 6: Two marmosets, M1 and M3, were placed facing each other; Figure 5 and Page 28: Two marmosets (M2 and M4) were placed face-to-face.

We apologize for the ambiguity; perhaps some of the confusion stems from the naming convention: M1, M2, M3, and M4 are the names of the marmosets. This has been clarified in the Methods. We have checked each analysis description that the marmoset names, number of runs, and number of sessions is stated, and have added the information is missing.

The marmoset animal model.

Imaging was performed on four common marmosets (named M1, M2, M3, and M4): 3-year-old males weighing 310 g (M1), 400 g (M2), and 340 g (M3), and a 2.5-year-old female weighing 365 g (M4).

Measuring the temporal signal-to-noise ratio.

Temporal SNR maps were calculated from a single-shot, EPI time series with two marmosets (M1 and M3) facing each other...

Assessing geometric distortion.

Marmosets M1 and M2 were placed facing each other 11-cm apart—a distance chosen based on the visual acuity of marmosets⁷. B_0 shimming was performed over a volume large enough to encompass both marmoset brains. Two single-shot, multiband⁸ EPI functional runs were acquired of both marmosets, simultaneously during a single session...

Intra-brain network mapping of constant interaction.

Four runs (during a single session), with 400 volumes each, were acquired with marmoset M1 and M3 facing each other and placed 11-cm apart.

Assessing socialization with in-person versus pre-recorded visible conspecific.

In paradigm 1, two marmosets (M3 and M4) were placed face-to-face...

In paradigm 2, M3 was replaced by a pre-recorded video of itself projected onto a screen located 11 cm in front of M4.

Each experiment consisted of 2 sessions, with 4 or 6 runs per session: the 10 total runs consisted of 5 runs with left-right acceleration and 5 runs with right-left acceleration.

Assessing coil sensitivity dependence on angular position. (Supplementary Methods)

Marmosets M3 and M4 were placed 11-cm apart and facing each other. A multi-slice, 3D gradient-recalled-echo image...

Transmit flip-angle mapping. (Supplementary Methods)

assess the difference in flip-angle between marmosets M3 and M4 when ...

Supplementary coil performance metrics. (Supplementary Methods)

Image SNR, receive sensitivity, and the geometry factor were measured as additional metrics to assess coil performance. For these analyses, marmosets M3 and M4 were placed 11-cm apart and facing each other, and a multi-slice, 3D gradient-recalled-echo image was acquired with and without RF transmission

R2.3. Familiarity modulates social influences. Were the marmosets familiar with each other? Housed together?

The familiarity between marmosets used for assessment of social interaction has now been added to the Methods as follows:

“Marmosets M1 and M3 (scanned for network mapping with a visible conspecific) were housed together and are twin brothers. Marmosets M3 and M4 (scanned for assessing socialization with in-person versus a pre-recorded visible conspecific) were housed separately and not familiar with each other.”

R2.4. The authors may want to add some results about estimated motion (mean, or derivative motion regressors) so that the reader can get an idea of the direction and amplitude of the movements allowed by the restraint system.

Head motion has now been estimated for each marmoset (M1 and M3) during the four, 10-min functional runs acquired for assessing intra-brain network mapping with a continual visible conspecific. Translation and rotation during any given run were less than 140 μm and 0.6°, respectively. This translation is a fraction of the 1-mm voxel size implemented in this study; therefore, motion had a negligible effect on data quality. Supplementary Fig. 1 now provides these results to illustrate the direction of movement.

Results:

“...four-point fixation of the chamber minimized translational and rotational motion to less than 140 μm and 0.6° over a 5-min functional run (Supplementary Fig. 1). Motion was considerably smaller than the 1-mm voxel size implemented in this study, thereby resulting in a negligible effect on data quality.”

Methods:

“**Assessing motion during functional runs.** Head motion was estimated for marmosets M1 and M3 during the four, 10-min functional runs acquired for assessing intra-brain network mapping with a continual visible conspecific. Each functional volume was registered to the middle volume of its respective time course (FSL; mflirt_acc). Time-varying motion parameters—translation (left-right, inferior-superior, posterior-anterior) and rotation (pitch, roll, yaw)—were then extracted for each run.”

Supplementary Fig. 1 | Estimated motion during functional time courses. Translational and rotational motion of marmosets M1 and M3 during four, 10-min functional runs. The social coil employs four-point fixation of a chamber, resulting in less than 140 μm of translation and 0.6° of rotation during a single run.

Major concern

R2.5. I do not think that the behavior measured qualifies as a "social interaction", let alone a "constant social interaction". The bore was alternatively blacked-out and illuminated during scanning. So brain activation was measured simultaneously in two animals facing either a visible or invisible conspecific. There was an online video control of whether or not the marmoset was awake, but no offline quantification of the time actually spent looking at the conspecific. Social interaction is the reciprocal influence subjects exert on each other. The two marmosets in the present study exert no influence on each other, and may or may not look at each other. So the affirmation repeated throughout the manuscript that the data provides a brain network mapping of constant social interaction should probably be tempered accordingly. What is detected is more modestly the brain network associated with a visible conspecific.

This is a fair concern and is echoed by Reviewer #3 and the editor. To address this concern, we have replaced the second experiment (intermittent interaction) with a new experiment that we believe demonstrates that the marmosets are indeed interacting—a comparison of the brain activation of a marmoset when either viewing a second marmoset in-person or when viewing a

pre-recorded video of the same marmoset—i.e., when either capable or incapable of socially interacting with a visible conspecific.

In the “constant interaction” experiment, we have now tempered the language, as the reviewer makes a salient point that we do not provide a quantification of the time marmosets actually spent looking at each other, so we should not refer to this as “constant interaction.” We now refer to this experiment as having a continually visible conspecific.

We have modified how the manuscript is framed: experiment 1 now refers to having a continually visible conspecific, while the (new) experiment 2—a comparison between an in-person versus pre-recorded video of a second conspecific—demonstrates that the marmosets are actually socially interacting.

Reviewer #3

This study describes a non-invasive fMRI method for studying brain activity in two marmosets simultaneously during ‘social interaction’ in a human 3T scanner. There have been previous studies of social interaction using simultaneous scanning (hyperscanning) of 2 or more people in single (e.g. dyads: Miyata et al Neuroimage Mar 2021, triads: Xie et al 2020 PNAS 117:23066) and multiple scanners (e.g. Read Montague). Multiple mice have been scanned in single MRI scanners, but not in the context of social interaction. To my knowledge, there are no publications of direct social interactions between animals in single scanners. This study scans two marmosets simultaneously as they view each other face to face. Marmosets live in families, exhibit characteristic social behaviors, are known especially for their auditory processing (Xiaoqing Wang), and social calls (Asif Ghazanfar). They have also become a subject of neuroscience studies, transgenic manipulation, and models of diseases such as autism (e.g. Mimura 2019 Neuroimage 195:243). Marmosets have thus become a popular nonhuman primate model of human behavior and disease. The combination of a strong nonhuman primate model and hyperscanning is thus a quite novel approach. The authors have developed a cutting edge technical advance in overcoming the many challenges of field inhomogeneity, RF coil design, and characterizing and decoupling noise in the two coils. This engineering is impressive and sets the stage for a new paradigm of studies in primate models of social interaction. While the authors are to be commended for such an advance, with respect to social interactions, it is not yet clear that this setup can be used for studying social interactions in a way that is more ‘social’ than viewing another marmoset on a screen. In my opinion, this is a great proof-of-principle that this approach can be achieved technically and I recognize what a tour-de-force effort this is. However, as presented, it may be more suitable for a journal targeted to new technologies or methods.

Dr. Roe,

Thank you very much for taking the time to provide a thoughtful and thorough review of our manuscript in which you raise a number of valid points. We have made substantial revisions to manuscript in response to these comments. Most importantly, we have replaced our second experiment (lights on/lights off intermittent interaction) with a new experiment that more definitively shows that the marmosets are interacting. This experiment compares the brain

activation of a marmoset when either viewing a second marmoset in-person or when viewing a pre-recorded video of the same marmoset—i.e., when either capable or incapable of socially interacting with a visible conspecific. When viewing a second marmoset in-person, activation was increased in regions associated with social cognition, thereby demonstrating that the marmosets are indeed interacting. We believe this has strengthened the manuscript and appreciate your guidance in this matter. Detailed responses to your comments, including a description of the new experiment and its results, are provided below.

Major comments

R3.1. To make this a more scientific question-based study, it would be helpful to offer some hypothesis or some predictions about which circuits social interaction would be activated by in-person interaction vs more passive monitor-based presentations (something quite relevant in this COVID world!). The data could then be probed to address such hypotheses.

As described directly above, we have now replaced our second experiment to demonstrate that the marmosets are indeed interacting and to discern which regions of the brain are preferentially activated due to this interaction, as detailed below. The following amendments have been made to the manuscript:

Abstract:

Notably, the brain activation of a marmoset when viewing a second marmoset in-person versus when viewing a pre-recorded video of the same marmoset—i.e., when either capable or incapable of socially interacting with a visible conspecific—demonstrated increased activity in regions of the brain responsible for processing social interaction.

Introduction:

In the first experiment, the method's efficacy is demonstrated by measuring the intra- and inter-brain activation of two marmosets who are continually within each other's visual field. In the second experiment, social interaction is demonstrated by comparing the brain activation of a marmoset when either viewing a second marmoset in-person or when viewing a pre-recorded video of the same marmoset—i.e., when either capable or incapable of socially interacting with a visible conspecific.

Results:

Assessing socialization with in-person versus pre-recorded conspecific. Brain activation maps of a marmoset were acquired when either viewing a second marmoset in-person (paradigm 1) or when viewing a pre-recorded video of the second marmoset (paradigm 2)—i.e., when either capable or incapable of socially interacting with a visible conspecific. In each of the two experiments, two marmosets (the second being in-person or a video) were placed face-to-face and separated by smart films, the opacity of which was alternated in a block design (Fig. 6a) to permit or deny visual contact; regions of increased brain activation were derived in each experiment by comparing the two epochs of the block design. For each paradigm, the stimulus condition (transparent smart films) was compared to the baseline condition (opaque smart films) using a two-sided paired t-test: resultant t-score maps discerned regions of the brain preferentially activated when the marmoset was viewing another marmoset (either face-to-face or face-to-video).

Activation patterns derived from paradigms 1 and 2 had similar spatial distributions, with significant activation occurring for both paradigms in multiple cortical areas (Fig. 6b): visual (V1, V2, V3, V4, V4T, V6A, V6, 19M), temporal (IPa, TE3, temporo-parieto-occipital association, fundus of the superior temporal, middle superior temporal, middle temporal), parietal (PFG, PG, PGM, occipito-parietal transitional area of cortex, anterior intraparietal, lateral intraparietal, medial intraparietal, ventral intraparietal) and frontal (dorsorostral and dorsocaudal parts of area 6, dorsal and ventral parts of area 8, caudal part of area 8, part a and b of the ventral area 6, part c of the primary motor area 4), and cingulate (23a, 24a, 24b, 24c, 24d, 25) cortex. At the subcortical level, significant activation was found across both paradigms in the putamen, caudate, amygdala, thalamus, superior colliculus, and cerebellum (Fig. 6c).

Viewing a second marmoset in-person produced both additional regions of activation, as well as more significant (higher t-values) and spatially expansive activation at both the cortical and subcortical levels. Additional regions of activation were found in multiple cortices: temporal (TE1), parietal (PE), frontal (parts a and b of the primary motor area 4), and somatosensory (1/2, 3a, 3b). The parietal, frontal, and temporal lobe showed higher significance levels in comparison to viewing a video of the second marmoset. At the subcortical level, the inferior colliculus was additionally activated, concurrent with higher significance levels in the caudate, putamen, amygdala, and thalamus.

Discussion:

Brain-activation maps generated of a marmoset when viewing a second marmoset in-person or when viewing a pre-recorded video of the second marmoset showed activation networks previously identified in marmosets performing tasks⁹⁻¹¹—most notably, this activation was found in regions associated with social cognition. An in-person conspecific resulted in additional regions of activation, as well as more significant and spatially expansive activation—this elucidates the difference in brain activation created by the ability to socially interact with a visible conspecific.

Activation linked to the visuo-saccadic network¹¹ (frontal, parietal, temporal, and visual regions of the marmoset brain) was observed in both paradigms, yet was stronger in the frontal- and parietal-related areas with an in-person conspecific, as a marmoset engaged in “real” visual contact will perform a greater number of saccades to analyze the other marmoset’s face¹². Most likely due to the same mechanism, regions typically activated during facial scanning^{9, 13} (i.e., face-patch regions in the temporal cortex—AD, MD, PD, PV—and in the anterior cingulate/frontal lateral cortex area 8/24) were present in both paradigm’s activation maps, yet significantly stronger for PD and PV patches with an in-person conspecific. Furthermore, with an in-person conspecific, activation was elevated and more expansive in areas of the brain that activate in response to light tactile stimulations to the face¹⁴: the lower part of somatosensory areas 3a, 3b, and 1/2 and bilateral motor areas. Despite the marmosets’ inability to make physical contact during scanning, when face-to-face their close proximity may have elicited an anticipation response to potential interaction or contact.¹⁵ The somatosensory and motor cortices have also been linked to the mirror system neuron¹⁶⁻¹⁹, theory of mind, emotions, and empathy in humans and non-human primates. Finally, a bilateral activation of area 25 and more expansive activation in the amygdala was observed with an in-person conspecific. Area 25 is known to be closely associated with emotions, visceromotor function, and memory²⁰⁻²³ and has structural connectivity with the amygdala: a recent fMRI study identified these areas as being involved in the observation of social interaction between marmosets¹⁵.

Differences between the brain-activation maps of the two paradigms are evident, demonstrating the efficacy of the social-coil method. With higher statistical power (i.e., more functional data), a statistical analysis between the two paradigms could be performed to further discriminate differences.

The method described in this study enables the measurement and assessment of synchronous neuronal activation, across the whole brain, between marmosets. In-person interaction between marmosets is shown to produce localized increases in brain activity compared to the presence of a visible, but non-interacting, marmoset—demonstrating the efficacy of the method in evaluating social interaction ...

Methods:

Assessing socialization with in-person versus pre-recorded visible conspecific. A marmoset’s brain activation was measured when either viewing another marmoset in-person (paradigm 1) or when viewing a pre-recorded video of the same marmoset (paradigm 2).

In paradigm 1, two marmosets (M3 and M4) were placed face-to-face, at a distance of 11 cm. Smart films (ASIN: B077P4QJT1; HOHOFILM) were secured approximately 2 cm in front of each marmoset. In a block paradigm, the opacity of the smart films was alternated by the application of a voltage: each run was comprised of 17 alternating blocks—18 s in the opaque condition (i.e., no visible conspecific) followed by 12 s in the transparent condition (i.e., capable of interacting with a visible conspecific). (Smart films were located close to the marmosets’ faces to prevent them from focusing on potential reflections off of the screen.) The voltage was controlled by a Raspberry-Pi (Raspberry-Pi 3, Model B), through a Python script, which was synced to the trigger output of the scanner that was sent at the beginning of each volume acquisition.

In paradigm 2, M3 was replaced by a pre-recorded video of itself projected onto a screen located 11 cm in front of M4. The video consisted of a recording of M3 while head-fixed in the receive coil and on the positioning platform. The brightness of the video display was adjusted to best replicate the light conditions

of the face-to-face paradigm. The opacity of the smart films was once again alternated to switch between the two conditions: no visible conspecific versus a visible conspecific without the ability to socially interact.

... Each experiment consisted of 2 sessions, with 4 or 6 runs per session: the 10 total runs consisted of 5 runs with left-right acceleration and 5 runs with right-left acceleration. ...

Fig. 6 | Preferentially activated brain regions during social interaction. **a**, Two task-based fMRI paradigms were acquired. In paradigm 1, a marmoset (M4) could view a second marmoset (M3) in-person—i.e., it was capable of socially interacting with a visible conspecific. In paradigm 2, M4 viewed a pre-recorded video of M3—i.e., it was incapable of socially interacting with a visible conspecific. In each paradigm, the two marmosets (M3 being in-person or a video) were placed face-to-face, 11-cm apart, and separated by smart films. The opacity of the smart films was alternated in a block design to permit or deny visual contact. **b**, Regions of increased brain activation were derived for each paradigm by comparing the two epochs of the block design in **a**. Activation maps are represented on the left and right fiducial surface of the M4 marmoset brain (t -scores > 2.26 , $p < 0.05$, uncorrected), with white lines representing cytoarchitectonic borders. **c**, The equivalent activation maps as represented on coronal slices to highlight subcortical activations. The displayed sagittal slice indicates the position of coronal slices.

R3.2. Social Interaction face-to-face vs viewing on monitor: Perhaps the most pertinent question is whether and to what extent the marmosets were socially interacting. Comparison of brain activity during viewing of another marmoset face on a monitor would be a reasonable comparison. Some controls would be helpful. These could take the form of devices

to track the animal's physiological and behavioral state, neurophysiological signals that correlate with attention (e.g. pupil size or other signatures), monitoring of the animals face and ears, tracking hormonal levels.

We believe the new second experiment, as detailed in the response to the reviewer's first question (R3.1) addresses this concern. The new experiment performs the face-to-face versus viewing-a-monitor comparison and compares the brain activity. Significant differences between the two paradigms existed, demonstrating the extent to which the marmosets were interacting.

At present, we do not have the equipment to monitor the physiological and behavioural state of two marmosets simultaneously. This is certainly something we would like to be able to do in the future.

R3.3. Social interaction vs vision: The primary control here is turning out the lights, during which it is assumed that the marmosets do not 'see' each other, thereby reducing the amount of social interaction. This comparison could also be interpreted as simply presence vs absence of visual inputs. Therefore, visual stimulus activations should also be presented to demonstrate that the differences observed are beyond simple vision.

This is a valid point. We replaced this experiment in response to the concerns expressed in R3.1 and R3.2. In this new experiment, we avoid the confounds of turning the light on and off (and the commensurate presence/absence of visual inputs) by using smart films between the animals. The opacity of the smart film is alternated by applying a voltage in sync with the scanner. This alters the animals' abilities to see each other without the cessation of visual input.

R3.4. Social interaction vs gender: Another useful comparison might be comparison of same sex and opposite sex brain activations. One would predict different activations based on gender.

Investigating sex-based differences in brain activation is an exciting avenue that our method should allow. This is certainly something we would like to investigate, but we believe this should be an entirely new study that would warrant its own manuscript. With the new experiments and data that were added to this manuscript in response to the reviewer's suggestions, R3.1 – R3.3, we believe we have now surpassed the proof-of-principle threshold and validated a question-based hypothesis.

R3.5. Marmoset behavior is strongly dominated by auditory inputs and their interpretation. Does the loud scanner noise lead to impairing of social interaction or to stress response?

The animals are acclimatized to the scanner environment, including the loud scanner noises, for a period of three weeks prior to scanning (this is now detailed in the response to R2.1): this minimizes their stress response while in the magnet.

The scanner noise certainly inhibits auditory communication between marmosets. Perhaps in the future we can circumvent this problem with some clever engineering, but at present it is not possible.

We have now added comments to the Methods section to address these questions:

“To minimize stress and anxiety during scanning, prior to the first imaging session marmosets were trained for three weeks following an acclimatization procedure^{3,4}. In week 1, marmosets were constrained in the restraint tube (Figure 1)—without head fixation—for durations increasing up to a maximum of 30 min. In week 2, the restraint tube was inserted into a mock MRI tube and gradient-coil sounds were played at increasing volume (up to ~80 dB) and for durations increasing up to a maximum of 60 min. In week 3, marmosets were head-fixed with the fixation pins (Figure 1), inserted into the mock MRI tube, and exposed to MRI sounds. In the current setup, auditory interaction between marmosets^{5,6} is inhibited by the loud noises of the MRI scanner; although this may prove to be a tractable problem through engineering, it presently limits this form of social interaction.

During each training session, animals were rewarded with pudding or marshmallow fluff for remaining calm, facing forward, and having minimal limb movement. Marmosets’ tolerance to training was assessed using a behavioral rating scale³: each marmoset was required to score a 1 or 2 on the assessment scale prior to scanning.”

R3.6. Social activity is typically accompanied by strong amygdala activation, as has been demonstrated in humans and macaque monkeys. Was this observed in the marmoset scans?

Yes. We observed strong amygdala activation in both paradigm 1 (face-to-face) and paradigm 2 (face-to-video) of the second experiment, with a more expansive activation for an in-person conspecific. This is notably linked with a strong bilateral activation of area 25—an area structurally connected to the amygdala and that is also strongly involved in emotion and social activity. Figure 6 illustrates the activation in the amygdala and area 25. The following comments have been added to the manuscript:

Results:

At the subcortical level, significant activation was found across both paradigms in the putamen, caudate, amygdala, thalamus, superior colliculus, and cerebellum (Fig. 6c).

Viewing a second marmoset in-person ... At the subcortical level, the inferior colliculus was additionally activated, concurrent with higher significance levels in the caudate, putamen, amygdala, and thalamus.

Discussion:

Finally, a bilateral activation of area 25 and more expansive activation in the amygdala was observed with an in-person conspecific. Area 25 is known to be closely associated with emotions, visceromotor function, and memory⁶⁰⁻⁶³ and has structural connectivity with the amygdala: a recent fMRI study identified these areas as being involved in the observation of social interaction between marmosets⁵⁵.

R3.7. References are not complete.

We have added the references cited by the reviewer(s). If there are additional references that the reviewer feels should be added, please let us know and we can include them. Here is a list of the added references:

1. Miller, C.T. et al. Marmosets: A neuroscientific model of human social behavior. *Neuron* **90**, 219-233 (2016).
2. Mimura, K. et al. Abnormal axon guidance signals and reduced interhemispheric connection via anterior commissure in neonates of marmoset asd model. *Neuroimage* **195**, 243-251 (2019).
3. Miyata, K. et al. Neural substrates for sharing intention in action during face-to-face imitation. *Neuroimage* **233**, 117916 (2021).

4. Takahashi, D.Y. et al. The developmental dynamics of marmoset monkey vocal production. *Science* **349**, 734-738 (2015).
5. Wang, X.Q., Merzenich, M.M., Beitel, R. & Schreiner, C.E. Representation of a species-specific vocalization in the primary auditory cortex of the common marmoset: Temporal and spectral characteristics. *Journal of Neurophysiology* **74**, 2685-2706 (1995).
6. Xie, H. et al. Finding the neural correlates of collaboration using a three-person fMRI hyperscanning paradigm. *Proceedings of the National Academy of Sciences* **117**, 23066-23072 (2020).

Minor comments

R3.8. Could this be done in a 9.4T? To what extent could this be conducted with commercial coils? Could you provide spatial SNR, g-factor, and the B1 maps/efficiency of two marmoset brains that are 11cm away from each other.

Whether this experiment could be performed on a 9.4T scanner is dependent on the size of the gradient coil and its region of linearity. As long as the marmosets can both be placed within the linear region of the gradient coil, and sufficiently far apart for their visual acuity, then this technique can translate to different field strengths and scanners. Our small-animal 9.4T magnet has a 15-cm inner diameter gradient coil; therefore, the region of linearity is too small along the z-axis to allow for such experiments. A statement has been added to the Methods section to address this comment:

“Whether an MRI scanner is capable of supporting this method is dependent on the size of the gradient coil: so as long as both marmosets can be placed within the linear region of the gradient coil, and sufficiently distant for their visual acuity, this technique can be translated to different field strengths and MRI scanners.”

Few commercial solutions exist for marmosets. Without any personal experience evaluating the few that do exist, we hesitate to speculate on whether or not they would be sufficient for this purpose. Ideally, they would need to restrain the animal sufficiently, allow for an unobstructed line-of-sight, and possess sufficient channel count for parallel acceleration.

We have now added flip-angle maps, image (spatial) SNR maps, B_1^- sensitivity maps of individual channels, and geometry factor maps to the Supplementary Methods, Discussion, and Figures.

Methods:

Supplementary coil performance metrics. Image SNR, receive sensitivity, and geometry factor were measured as supplementary metrics to assess coil performance. A full description of this analysis is provided in the Supplementary Methods and Discussion.

Measuring the temporal signal-to-noise ratio. ... Spatial variations in SNR due to flip-angle inhomogeneity were found to be minimal, as described in the Supplementary Methods and Discussion.

Supplementary Methods:

Transmit flip-angle mapping. The transmit-field uniformity, as produced by the scanner’s body coil, was measured to assess the difference in flip-angle between marmosets M3 and M4 when placed 11-cm apart and facing each other. Flip-angle maps were measured using a turbo, fast-low-angle-shot pulse sequence: matrix size: 256 × 80, FOV: 240 × 75 mm, number of slices: 12, slice thickness: 2 mm, TE/TR: 2.6/6,000 ms, flip angle: 8°, BW: 490 Hz/pixel. The mean flip angle over each head was computed in Matlab.

Supplementary coil performance metrics. Image SNR, receive sensitivity, and the geometry factor were measured as additional metrics to assess coil performance. For these analyses, marmosets M3 and M4 were placed 11-cm apart and facing each other, and a multi-slice, 3D gradient-recalled-echo image was acquired

with and without RF transmission (matrix size: $288 \times 104 \times 64$, FOV: $220 \times 79 \times 51.2$ mm, TE/TR: 4.4/10 ms, flip angle: 20° , BW: 220 Hz/pixel, number of averages: 2).

Image SNR maps were derived from the complex images of individual elements using a covariance-weighted, root-sum-of-squares reconstruction^{1,2}. The constituent sensitivity profiles of receive elements were calculated by dividing the image of each receiver by the combined image.

The geometry factor was calculated by retrospectively under-sampling k-space by two-fold in the left-right direction—i.e., replicating the acceleration factor and field-of-view employed in the acquisition of functional images in this study. The worst-case geometry-factor was estimated by cropping the fully sampled images tight to the head prior to calculating the geometry factor in the left-right, superior-inferior, and anterior-posterior directions. Inverse geometry-factor maps were reconstructed in Matlab using the sensitivity encoding (SENSE) method²⁴.

Supplementary Discussion:

The social-coil method does not rely on a particular transmit coil; however, the transmit coil must be of sufficient size to produce a uniform and consistent flip angle across both marmosets, thereby obviating a commensurate variance in spatial and temporal SNR. Flip-angle maps acquired with the scanner's body coil (Supplementary Fig. 4) showed only a small difference in mean flip angle (5.8%) between marmosets M3 and M4 when placed 11-cm apart.

In vivo image SNR maps showed high SNR in the peripheral cortex, as expected for a surface coil array (Supplementary Fig. 5a). The difference in image SNR between marmoset M3 (in coil 1) and marmoset M4 (in coil 2) was only 5% in the centre of the brain and 1% in the peripheral cortex. These differences are due to minor discrepancies between receive-array construction and between marmoset anatomies. The constituent sensitivity profiles of receive elements (Supplementary Fig. 5b) are spatially independent, which reduces the noise amplification during the reconstruction of accelerated images (i.e., the geometry factor).

The geometry factor was equal to unity throughout the brain when accelerating two-fold in the left-right direction with a field-of-view equivalent to that of functional acquisitions in this study. This was a result of the field-of-view in the left-right phase-encode direction being twice that of a marmoset's head width; therefore, image replicas created by under-sampling k-space did not overlap. After cropping images (prior to retrospective under-sampling) to ensure the maximum possible overlap of image replicas (i.e., the worst-case noise amplification), the mean/maximum geometry factor in the left-right, superior-inferior, and anterior-posterior directions was 1.30/1.79, 1.24/1.62, and 1.47/2.41, respectively (Supplementary Fig. 5c).

Supplementary Fig. 4 | Transmit flip-angle maps. A representative axial slice of a transmit flip-angle map acquired of marmosets M3 and M4 when placed 11-cm apart. The confluence of a large-diameter transmit body coil and the relatively small marmoset head produces a consistent flip angle between coils: the relative difference between mean flip angle of the phantoms differed by 5.8%. Flip-angle maps have been reoriented in radiological convention.

Supplementary Fig. 5 | Coil performance metrics. **a**, Image SNR maps, together with temporal SNR maps (Fig. 2), can be used to assess the similarity in performance between the two receive arrays. Representative sagittal, axial, and coronal slices show similar image SNR profiles between coils. Image SNR differs between the two receive array/marmoset combinations by 5% in the centre of the brain and 1% in the peripheral cortex (as depicted by the dashed ROIs). **b**, Receive sensitivity maps of individual receive elements (corresponding to the planar coil layout) govern the noise amplification during parallel-imaging reconstruction. Each receive array of the social-coil method is comprised of five elements—four laterally and one at the superior aspect of the head—thereby making it capable of a two-fold acceleration rate in each direction. **c**, Inverse geometry-factor maps in the anterior-posterior (A-P), left-right (L-R), and superior-inferior directions (S-I) were calculated with a two-fold acceleration rate and with images cropped to induce the worst-case geometry factor. The mean and maximum geometry factor are provided below individual maps.

R3.9. To highlight the advantages of this approach, some comparison of this study with those of e.g. multichannel electrophysiological recordings and of miniature scope imaging would be useful. What are the similarities and differences in results and in approach?

Brain activation of socially interacting marmosets has not been studied previously, which adds to the uniqueness of this manuscript. That being said, we have now augmented our comparison of the results and approaches of complementary techniques (electrophysiological recordings, calcium imaging, and FDG-PET) when studying macaque monkeys, mice, and bats.

When referring to the *results* of these studies, we now name the technique explicitly (if not previously included) to allow direct comparison:

“...neuronal ensemble recordings of non-human primates have shown inter-brain cortical synchronization during social interaction²⁵.”

“...the mere presence of another monkey during the completion of a task has been demonstrated to increase brain activity in the attention frontoparietal network **using FDG-PET**²⁶.”

“...these brain regions in macaque monkeys have shown increased activity **(using BOLD-based functional MRI)** when they watch movies showing monkeys interacting versus acting independently²⁷.”

“...calcium imaging of socially interacting mice has demonstrated synchrony of their neural activity predictive of social behaviour²⁸; wireless electrophysiology used to record local field potentials of socially interacting bats has demonstrated correlation of neural activity over a range of timescales²⁹.”

The primary advantage of fMRI in terms of *approach* is that it is a whole-brain technique, in contrast to electrophysiological recordings and calcium imaging, albeit at a lower spatial resolution than these techniques. The advantage of being a whole-brain technique has been cited in the Introduction and Discussion; however, we now include a direct comparison to electrophysiological recordings and calcium imaging:

“...the study of socially interacting animals has yet to be investigated with fMRI—a technique which would allow a whole-brain assessment of activation (in contrast to higher spatial-resolution electrophysiological recordings and calcium imaging) in multiple animals simultaneously.”

References

1. Roemer, P.B., Edelstein, W.A., Hayes, C.E., Souza, S.P. & Mueller, O.M. The NMR phased-array. *Magnetic Resonance in Medicine* **16**, 192-225 (1990).
2. Kellman, P. & McVeigh, E.R. Image reconstruction in SNR units: A general method for SNR measurement. *Magnetic Resonance in Medicine* **54**, 1439-1447 (2005).
3. Silva, A.C. et al. in *Magnetic resonance neuroimaging: Methods and protocols.* (eds. M. Modo & J.W.M. Bulte) 281-302 (Humana Press, Totowa, NJ; 2011).
4. Schaeffer, D.J. et al. Integrated radiofrequency array and animal holder design for minimizing head motion during awake marmoset functional magnetic resonance imaging. *Neuroimage* **193**, 126-138 (2019).
5. Wang, X.Q., Merzenich, M.M., Beitel, R. & Schreiner, C.E. Representation of a species-specific vocalization in the primary auditory cortex of the common marmoset: Temporal and spectral characteristics. *Journal of Neurophysiology* **74**, 2685-2706 (1995).
6. Takahashi, D.Y. et al. The developmental dynamics of marmoset monkey vocal production. *Science* **349**, 734-738 (2015).
7. Nummela, S.U. et al. Psychophysical measurement of marmoset acuity and myopia. *Developmental Neurobiology* **77**, 300-313 (2017).
8. Moeller, S. et al. Multiband multislice GE-EPI at 7 Tesla, with 16-fold acceleration using partial parallel imaging with application to high spatial and temporal whole-brain fMRI. *Magnetic Resonance in Medicine* **63**, 1144-1153 (2010).
9. Hung, C.-C. et al. Functional MRI of visual responses in the awake, behaving marmoset. *Neuroimage* **120**, 1-11 (2015).
10. Cléry, J.C. et al. Looming and receding visual networks in awake marmosets investigated with fMRI. *Neuroimage* **215** (2020).
11. Schaeffer, D.J. et al. Task-based fMRI of a free-viewing visuo-saccadic network in the marmoset monkey. *Neuroimage* **202** (2019).
12. Mitchell, J.F., Reynolds, J.H. & Miller, C.T. Active vision in marmosets: A model system for visual neuroscience. *Journal of Neuroscience* **34**, 1183-1194 (2014).
13. Schaeffer, D.J. et al. Face selective patches in marmoset frontal cortex. *Nature Communications* **11** (2020).
14. Cléry, J.C. et al. Whole brain mapping of somatosensory responses in awake marmosets investigated with ultra-high-field fMRI. *Journal of Neurophysiology* **124**, 1900-1913 (2020).

15. Clery, J.C., Hori, Y., Schaeffer, D.J., Menon, R.S. & Everling, S. Neural network of social interaction observation in marmosets. *Elife* **10** (2021).
16. Rajmohan, V. & Mohandas, E. Mirror neuron system. *Indian Journal of Psychiatry* **49**, 66-69 (2007).
17. Bastiaansen, J., Thioux, M. & Keysers, C. Evidence for mirror systems in emotions. *Philosophical Transactions of the Royal Society B-Biological Sciences* **364**, 2391-2404 (2009).
18. Keysers, C., Kaas, J.H. & Gazzola, V. Somatosensation in social perception. *Nature Reviews Neuroscience* **11**, 417-428 (2010).
19. Amodio, D.M. & Keysers, C. Editorial overview: New advances in social neuroscience: From neural computations to social structures. *Current Opinion in Psychology* **24**, IV-VI (2018).
20. Alexander, L., Clarke, H.F. & Roberts, A.C. A focus on the functions of area 25. *Brain Sciences* **9**, 129 (2019).
21. Alexander, L. et al. Over-activation of primate subgenual cingulate cortex enhances the cardiovascular, behavioral and neural responses to threat. *Nature Communications* **11**, 5386 (2020).
22. Joyce, M.K.P. & Barbas, H. Cortical connections position primate area 25 as a keystone for interoception, emotion, and memory. *J. Neurosci.* **38**, 1677-1698 (2018).
23. Öngür, D., Ferry, A.T. & Price, J.L. Architectonic subdivision of the human orbital and medial prefrontal cortex. *Journal of Comparative Neurology* **460**, 425-449 (2003).
24. Pruessmann, K.P., Weiger, M., Scheidegger, M.B. & Boesiger, P. SENSE: Sensitivity encoding for fast MRI. *Magnetic Resonance in Medicine* **42**, 952-962 (1999).
25. Tseng, P.-H., Rajangam, S., Lehew, G., Lebedev, M.A. & Nicolelis, M.A.L. Interbrain cortical synchronization encodes multiple aspects of social interactions in monkey pairs. *Sci. Rep.* **8** (2018).
26. Monfardini, E. et al. Others' sheer presence boosts brain activity in the attention (but not the motivation) network. *Cerebral Cortex* **26**, 2427-2439 (2015).
27. Sliwa, J. & Freiwald, W.A. A dedicated network for social interaction processing in the primate brain. *Science* **356**, 745-+ (2017).
28. Kingsbury, L. et al. Correlated neural activity and encoding of behavior across brains of socially interacting animals. *Cell* **178**, 429-+ (2019).
29. Zhang, W.J. & Yartsev, M.M. Correlated neural activity across the brains of socially interacting bats. *Cell* **178**, 413-+ (2019).

REVIEWER COMMENTS

Reviewer #1 (Remarks to the Author):

I thank the authors for their replies to my questions. I fully support the publication of their improved revised manuscript.

Reviewer #2 (Remarks to the Author):

The authors have taken each request into consideration in their revised version and have added the requested additional data. These corrections together with the addition of a new experiment allowing comparison of actual versus virtual social presence considerably improve the manuscript. I think that this is an excellent paper ready to be published.

Martine Meunier, Lyon, France

Reviewer #3 (Remarks to the Author):

This manuscript reports a technological advance of recording from two awake marmosets within the same MRI. This is novel. No other study has demonstrated the ability to image two awake monkeys, face to face, simultaneously within a single MRI. The study demonstrates brain responses to images or videos of other marmosets in social situations. The activated brain areas include visual and some limbic regions, consistent with previous studies in monkeys and humans. However, the claim that this demonstrates social interaction between the two monkeys is not supported. I have reservations, given my experience with marmoset behavior and the published studies from Wang Xiaoqing's lab, that marmosets will behave socially in such an environment. That is not to say it is impossible, just not yet demonstrated.

What I want to see is a direct comparison of the face-to-face with the face-to-video. It is not clear to me that they did this, as Paradigm 1 was social vs blank and Paradigm 2 was social video vs blank. Does the following indicate they compared social 1 vs social 2? "Marmoset t-value maps were then compared at the individual level using a two-sided paired t test (AFNI; 3dttest++); resultant t-values were displayed on fiducial maps using Connectome Workbench in conjunction with the NIH marmoset brain template." What are they pairing in the t-test? Does this mean pairing the same voxel under the 2 different conditions? pairing ALL the voxels? they do not specify which voxels. I did not see enough detail to evaluate what was done.

In fig6, the contrast is reasonably made by blocking and unblocking monkey's vision. Although the activity is very similar between seeing real animal and seeing video, this result could be due to setting low T-score cutoff. The result is that of separating the monkey brain into the visual part and the non-visual part. It still falls short of demonstrating social activity. If they can demonstrate monkeys talking to each other or anything that shows an interaction between two marmosets (i.e. a behavior in one marmoset that then leads to a behavior in the other marmoset), it would be more interesting. Or perhaps at least they could compare marmosets viewing a social condition vs a non-social condition.

I would love to see this work. However, given the many years of study by Wang Xiaoqing's lab of marmosets simply sitting without making calls or active interaction while headfixed, I am still not convinced that one can get head restrained marmosets to be social in the MRI.

Response to Reviews

We would like to thank the reviewers and editor once again for taking the time to provide thoughtful feedback on our manuscript. We have performed additional experiments and made amendments to the manuscript to address these concerns.

Two additional authors have been added to the manuscript, as they performed and analyzed the new eye-tracking experiment.

Reviewer #3:

R3.1. This manuscript reports a technological advance of recording from two awake marmosets within the same MRI. This is novel. No other study has demonstrated the ability to image two awake monkeys, face to face, simultaneously within a single MRI. The study demonstrates brain responses to images or videos of other marmosets in social situations. The activated brain areas include visual and some limbic regions, consistent with previous studies in monkeys and humans. However, the claim that this demonstrates social interaction between the two monkeys is not supported. I have reservations, given my experience with marmoset behavior and the published studies from Wang Xiaoqing's lab, that marmosets will behave socially in such an environment. That is not to say it is impossible, just not yet demonstrated.

We have made amendments to the manuscript to demonstrate mutual gaze between the two marmosets—a form of behavioural modification during social interaction—while head-fixed in the social coil. This was performed with simultaneous eye-tracking of the marmosets. Details of the experiment are provided in response to R3.3.

R3.2. What I want to see is a direct comparison of the face-to-face with the face-to-video. It is not clear to me that they did this, as Paradigm 1 was social vs blank and Paradigm 2 was social video vs blank. Does the following indicate they compared social 1 vs social 2? "Marmoset t-value maps were then compared at the individual level using a two-sided paired t test (AFNI; 3dttest++); resultant t-values were displayed on fiducial maps using Connectome Workbench in conjunction with the NIH marmoset brain template." What are they pairing in the t-test? Does this mean pairing the same voxel under the 2 different conditions? pairing ALL the voxels? they do not specify which voxels. I did not see enough detail to evaluate what was done.

We have now performed a direct comparison between the face-to-face and face-to-video paradigms. We performed an unpaired, voxel-wise t-test between the two stimulus conditions—i.e., face-to-face with transparent smart films versus face-to-video with transparent smart films. This has been clarified in the text and the resultant z-score map has been included as a subpanel in Figure 6. We have modified (pared down) the Results and Discussion sections to only discuss the

regions that met the threshold of significance in the t-test (notably the face-patch network), and modified a sentence in the Abstract to the same effect.

Abstract:

“Notably, the brain activation of a marmoset when viewing a second marmoset in-person versus when viewing a pre-recorded video of the same marmoset—i.e., when either capable or incapable of socially interacting with a visible conspecific—demonstrated increased activation in the face-patch network.”

Results:

“For each paradigm, the stimulus condition (transparent smart films) was compared to the baseline condition (opaque smart films) using a two-sided paired t-test (Fig. 6b, c); an unpaired t-test between the two paradigms’ stimulus conditions discerned regions of the brain preferentially activated when the marmoset was viewing another marmoset (face-to-face versus face-to-video; Fig. 6d).”

“Viewing a second marmoset in-person (paradigm 1) compared to a video (paradigm 2) resulted in significantly higher activations in regions corresponding to previously described temporal (AD, MD, PV, and PD) and frontal (area 8b/24) face patches^{1, 2}. In addition, we observed higher activations in the right motor and premotor areas (area 4, area 6DC, are 6DR, 8C) (Fig. 6d).”

Discussion:

“Brain-activation maps generated of a marmoset when viewing a second marmoset in-person or when viewing a pre-recorded video of the second marmoset showed activation networks previously identified in marmosets performing tasks³⁻⁵—most notably, this activation was found in regions associated with visuomotor tasks. An in-person conspecific resulted in significantly higher activations in the anterior cingulate (area 8B, area 24) and temporal regions. These areas correspond to previously identified frontal² and temporal face patches in marmosets^{1, 2}, supporting a role of the face patch network in social cognition⁶. Overall, differences between the brain-activation maps of the two paradigms are consistent with behavioural evidence of mutual gaze (see Supplementary Methods and Discussion). In humans, it has recently been suggested that eye contact during conversation indicates shared attention⁷.”

Methods:

“A direct comparison between the two paradigms was conducted at the individual level by performing an unpaired, voxel-wise t-test between the two paradigms’ stimulus conditions (AFNI; 3dttest++)—i.e., face-to-face with transparent smart films versus face-to-video with transparent smart films. Resultant z-values were displayed on fiducial maps using Connectome Workbench⁸ in conjunction with the NIH marmoset brain template⁹.”

Fig. 6 | Preferentially activated brain regions during social interaction.

... d, A voxel-wise t-test between the stimulus conditions of paradigms 1 and 2 indicates regions of increased activation due to socialization. ...

R.3.3. In fig6, the contrast is reasonably made by blocking and unblocking monkey's vision. Although the activity is very similar between seeing real animal and seeing video, this result could be due to setting low T-score cutoff. The result is that of separating the monkey brain into the visual part and the non-visual part. It still falls short of demonstrating social activity. If they can demonstrate monkeys talking to each other or anything that shows an interaction between two marmosets (i.e. a behavior in one marmoset that then leads to a behavior in the other marmoset), it would be more interesting. Or perhaps at least they could compare marmosets viewing a social condition vs a non-social condition.

I would love to see this work. However, given the many years of study by Wang Xiaoqing's lab of marmosets simply sitting without making calls or active interaction while headfixed, I am still not convinced that one can get head restrained marmosets to be social in the MRI.

In light of the reviewer's comments, we performed an experiment where we placed two marmosets in the social coil to check for vocalization. Prior to head-fixing, the animals would be quite vocal. After head-fixing, as the reviewer suggests, the marmosets stop their verbal calls. Although this precludes this form of interaction, head-fixation is nonetheless a requirement to perform MRI. We have now included a more definitive statement in the text in this regard.

That being said, the marmosets still perform mutual eye gazing—a behaviour indicative of social interaction. To demonstrate this, we performed simultaneous eye tracking of two marmosets on the bench, while head-fixed in the social coil with MRI sounds being played through a speaker. Marmosets demonstrated periods (approximately 33% of the time) in which they would look at the other marmoset's face. When time shifting these gaze patterns, we see a temporal correlation. The correlation is lower for a shift of about -1 s and higher for a shift of about +1 s. This means marmoset A would look at the face of marmoset B and a second later marmoset B would look at

marmoset A and a second later they break mutual gaze. This is consistent with prior findings in humans engaged in conversation, where ~2 s of gaze contact was reported¹⁰. It has also recently been reported that eye contact during conversation can indicate shared attention in humans⁷.

The following sections were added to the manuscript:

Discussion:

“Overall, differences between the brain-activation maps of the two paradigms are consistent with behavioural evidence of mutual gaze (see Supplementary Methods and Discussion). In humans, it has recently been suggested that eye contact during conversation indicates shared attention⁷.”

Methods:

“Benchtop eye-tracking experiments were performed between marmoset M4 and M5 (a 4-year-old female weighing 490 g). Marmosets M4 and M5 were housed separately and not familiar with each other.”

“Marmosets did not vocalize with each other^{11, 12} while head-fixed in the social coil; although head-fixation inhibits vocal communication, it is nonetheless a requisite to successfully perform MRI. Marmosets did, however, exhibit behaviour indicative of social interaction—i.e., mutual eye gaze. This was validated through simultaneous eye tracking of two head-fixed marmosets while on the bench (see Supplementary Methods and Discussion).”

Supplementary Methods:

“Eye movements during social interaction. To investigate the effects of social interactions on potentially linked patterns of eye movements between marmosets, we carried out eye movement recordings outside the scanner in an experimental setup approximating that used during MRI scanning. Marmosets M4 and M5 were head-fixed within the social coil and positioned facing each other at a viewing distance of 11 cm. At this distance, the face of the conspecific animal subtended a visual angle of approximately 20°. Eye movements were recorded at 500 Hz using video-oculography (EyeLink II, SR Research, Ottawa, ON, Canada). Separate cameras were placed in front of each animal out of the direct line-of-view of the conspecific animal.

To calibrate eye movements, marmosets viewed light-emitting diodes (LEDs), placed at known eccentricities, in the dark. A screen subtending 90° of visual angle was placed in front of the animals. Small LEDs (3.4°) were placed within the screen at an eccentricity of 24° to the left and right of centre and illuminated for a duration of 250 ms at an interstimulus interval of 3 s. The abrupt onset of these stimuli typically evoked saccades from the animals. This allowed the correlation between the voltage output of the eye tracker and the known eccentricity of the stimuli to be calibrated. The positions of the calibration stimuli were chosen such that they would roughly bracket the position of the conspecific animals' face. This procedure was carried out for each animal separately.

Following calibration, the room was re-illuminated, the screen was removed, and eye movements were recorded while the animals viewed each other for a duration of 15 minutes. Audio recordings of scanner sequences were played to simulate the MRI scanner environment.

Eye movement data was analyzed using custom python scripts. Calibrated eye traces were smoothed via linear convolution of an 11-sample-wide hamming window. A 20° square window was defined around the face of the conspecific animal's face; for each animal, it was determined when the animal's gaze was within this window. The degree of overlap between these time courses was then computed, with the process being subsequently repeated while shifting the time course of one animal forward or backward in time by one sample for up to 3 s.”

Supplementary Discussion:

“Simultaneous eye tracking of two marmosets demonstrated that marmosets engaged in intermittent mutual eye gaze—i.e., a behaviour indicative of social interaction (Supplementary Fig. 3). The gaze time courses of

marmosets M4 and M5 had peak overlap values at negative lags and lower overlap values at positive lags. This indicates marmoset M4 would look at the face of marmoset M5, followed by marmoset M5 looking at the face of marmoset M4, then the marmosets would break their mutual gaze. A similar pattern has been found in human-to-human gaze interactions and has been termed “antiphase synchrony”¹⁰.

Supplementary Fig. 3 | Synchrony between marmoset gaze patterns. **a**, The horizontal gaze position, E_h , of two marmosets (M4 and M5) when viewing each other while head-fixed in the social coil. A representative 60-s excerpt is shown of the 15-minute recording session. Time spent looking at the conspecific marmoset’s face is highlighted in grey for each marmoset and the overlap in this time between marmosets is highlighted in red. **b**, The proportion of time when gaze overlapped was computed at time lags ranging from -3 s to +3 s for the entire recording session. This indicates marmoset M4 would look at the face of marmoset M5, followed by marmoset M5 looking at the face of marmoset M4, then the marmosets would break their mutual gaze.

References

1. Hung, C.-C. et al. Functional mapping of face-selective regions in the extrastriate visual cortex of the marmoset. *The Journal of Neuroscience* **35**, 1160-1172 (2015).
2. Schaeffer, D.J. et al. Face selective patches in marmoset frontal cortex. *Nature Communications* **11** (2020).
3. Hung, C.-C. et al. Functional MRI of visual responses in the awake, behaving marmoset. *Neuroimage* **120**, 1-11 (2015).
4. Clery, J.C. et al. Looming and receding visual networks in awake marmosets investigated with fMRI. *Neuroimage* **215** (2020).
5. Schaeffer, D.J. et al. Task-based fMRI of a free-viewing visuo-saccadic network in the marmoset monkey. *Neuroimage* **202** (2019).

6. Schwiedrzik, C.M., Zarco, W., Everling, S. & Freiwald, W.A. Face patch resting state networks link face processing to social cognition. *PLoS Biol* **13**, e1002245 (2015).
7. Wohltjen, S. & Wheatley, T. Eye contact marks the rise and fall of shared attention in conversation. *Proc Natl Acad Sci U S A* **118** (2021).
8. Marcus, D. et al. Informatics and data mining tools and strategies for the human connectome project. *Frontiers in Neuroinformatics* **5** (2011).
9. Liu, C. et al. Marmoset brain mapping v3: Population multi-modal standard volumetric and surface-based templates. *Neuroimage* **226**, 117620 (2021).
10. Tschacher, W., Tschacher, N. & Stukenbrock, A. Eye synchrony: A method to capture mutual and joint attention in social eye movements. *Nonlinear Dynamics Psychol Life Sci* **25**, 309-333 (2021).
11. Wang, X.Q., Merzenich, M.M., Beitel, R. & Schreiner, C.E. Representation of a species-specific vocalization in the primary auditory cortex of the common marmoset: Temporal and spectral characteristics. *Journal of Neurophysiology* **74**, 2685-2706 (1995).
12. Takahashi, D.Y. et al. The developmental dynamics of marmoset monkey vocal production. *Science* **349**, 734-738 (2015).

REVIEWERS' COMMENTS

Reviewer #3 (Remarks to the Author):

The evidence supporting social interaction this is still rather weak. But the authors have done their best to show the possibility of a certain level of social interaction. 2 methods were tried.

- In Fig. 6, they tried to compare the MR signals during stimulus block of paradigm 1 and 2. This is not an ideal comparison, but it's technically difficult to alternate between real monkey and monkey video in a single block. From this comparison, they found face patches had higher activity when seeing real monkey, but the reproducibility of this comparison can be questioned.

- The mutual gaze figure (supp. fig. 3b) is weak. It's 36% vs. 35.5% of mutual gaze when comparing 1-sec and 0-sec gaze delay. I believe 1 monkey's gaze may lead to another. But I am not convinced with their supp. fig. 3b. Negative controls can be added to this figure by randomly shuffling the eye tracking data (e.g. 100 times).

There are two aspects of this setup that tend to inhibit social behavior in marmosets. One is head fixation, something known to suppress vocalizations and social behavior; and the other is the loud MRI environment. It remains to be tested whether and to what degree the marmosets can exhibit social behavior.

Given these considerations, I believe this is still something worth pursuing.

Response to Reviews

We would like to thank the reviewer and editor for once again taking the time to provide thoughtful feedback on our manuscript. We have performed the negative control for the eye-tracking experiment, as suggested by the reviewer, and amended the manuscript in accordance with the editor's comments.

Reviewer #3:

R3.1. The evidence supporting social interaction this is still rather weak. But the authors have done their best to show the possibility of a certain level of social interaction. 2 methods were tried.

- In Fig. 6, they tried to compare the MR signals during stimulus block of paradigm 1 and 2. This is not an ideal comparison, but it's technically difficult to alternate between real monkey and monkey video in a single block. From this comparison, they found face patches had higher activity when seeing real monkey, but the reproducibility of this comparison can be questioned.

- The mutual gaze figure (supp. fig. 3b) is weak. It's 36% vs. 35.5% of mutual gaze when comparing 1-sec and 0-sec gaze delay. I believe 1 monkey's gaze may lead to another. But I am not convinced with their supp. fig. 3b. Negative controls can be added to this figure by randomly shuffling the eye tracking data (e.g. 100 times).

There are two aspects of this setup that tend to inhibit social behavior in marmosets. One is head fixation, something known to suppress vocalizations and social behavior; and the other is the loud MRI environment. It remains to be tested whether and to what degree the marmosets can exhibit social behavior.

Given these considerations, I believe this is still something worth pursuing.

As the reviewer notes, verbal communication is limited within the scanner. We believe the eye-tracking data does illustrate the mutual eye gaze, though. Nevertheless, the editor has suggested to change the title of the manuscript, we assume to focus on the simultaneous fMRI of two marmosets with a visual conspecific (i.e., downplaying the 'social interaction'): we believe we have demonstrated this in the manuscript and are pleased with this outcome.

We have addressed the remaining suggestion of the reviewer—to calculate a negative control for the eye-tracking experiment. This has now been included in Supplementary Figure 3 and described in the Supplementary Methods.

Supplementary Methods:

The degree of overlap between these time courses was then computed, with the process being subsequently repeated while shifting the time course of one animal forward or backward in time by one sample for up to 5 s. A null distribution was computed by randomly shuffling the eye position of one animal in time before computing the degree of overlap. This process was repeated 1,000 times and the 2.5th and 97.5th percentiles were determined at each time lag.

Supplementary Figure 3:

A null distribution (grey line; range: 2.5th – 97.5th percentile; N: 1,000) was constructed by shuffling the eye position of one animal before computing the proportion of overlap at each time lag.